# Daily electrical activity in the master circadian clock of a diurnal mammal

**Beatriz Bano-Otalora[1,2†], Matthew J Moye[3,4†], Timothy Brown[1,5], Robert J Lucas[1,2*], Casey O Diekman[3,6*], Mino DC Belle[7*]**

[1]Centre for Biological Timing, Faculty of Biology Medicine and Health, University of Manchester, Manchester, United Kingdom; [2]Division of Neuroscience and Experimental Psychology, Faculty of Biology Medicine and Health, University of Manchester, Manchester, United Kingdom; [3]Department of Mathematical Sciences, New Jersey Institute of Technology, Newark, United States; [4]Department of Quantitative Pharmacology and Pharmacometrics (QP2), Kenilworth, United States; [5]Division of Diabetes, Endocrinology and Gastroenterology, Faculty of Biology Medicine and Health, University of Manchester, Manchester, United Kingdom; [6]EPSRC Centre for Predictive Modelling in Healthcare, Living Systems Institute, University of Exeter, Exeter, United Kingdom; [7]Institute of Biomedical and Clinical Sciences, University of Exeter Medical School, University of Exeter, Exeter, United Kingdom

**Abstract** Circadian rhythms in mammals are orchestrated by a central clock within the suprachiasmatic nuclei (SCN). Our understanding of the electrophysiological basis of SCN activity comes overwhelmingly from a small number of nocturnal rodent species, and the extent to which these are retained in day-active animals remains unclear. Here, we recorded the spontaneous and evoked electrical activity of single SCN neurons in the diurnal rodent *Rhabdomys pumilio*, and developed cutting-edge data assimilation and mathematical modeling approaches to uncover the underlying ionic mechanisms. As in nocturnal rodents, *R. pumilio* SCN neurons were more excited during daytime hours. By contrast, the evoked activity of *R. pumilio* neurons included a prominent suppressive response that is not present in the SCN of nocturnal rodents. Our modeling revealed and subsequent experiments confirmed transient subthreshold A-type potassium channels as the primary determinant of this response, and suggest a key role for this ionic mechanism in optimizing SCN function to accommodate *R. pumilio*'s diurnal niche.

**\*For correspondence:**
robert.lucas@manchester.ac.uk (RJL);
diekman@njit.edu (COD);
M.D.C.Belle@exeter.ac.uk (MDCB)

[†]These authors contributed equally to this work

## Introduction

The mammalian master circadian clock is localized within the hypothalamic suprachiasmatic nucleus (SCN), where nearly 20,000 neurons synchronize their daily activity with the light-dark cycle to orchestrate circadian rhythms in physiology and behavior (*Reppert and Weaver, 2002*). SCN neurons are electrically and chemically heterogeneous. Most, if not all, SCN neurons contain an internal molecular clock that operates on a transcription-translation feedback loop (TTFL) (*Ko and Takahashi, 2006*). Activity of the TTFL drives circadian rhythms in electrical activity, with SCN neurons notably more active during the day (up-state) than at night (down-state). This excitability landscape within the SCN is reinforced by the appropriate synaptic integration of extrinsic signals, which includes photic information from the retina and behavioral feedback reflecting arousal state (*Belle and Diekman, 2018*), endowing plasticity to the circadian timekeeping system (*Michel and Meijer, 2020*).

Our current understanding of SCN neurophysiology comes overwhelmingly from electrophysiological recordings on a small number of nocturnal rodent species (mice, rats, and hamsters)

(*Colwell, 2011*; *Belle and Diekman, 2018*; *Harvey et al., 2020*). A handful of studies have confirmed that the daytime peak in spontaneous action potential firing activity (as reflected in extracellular electrical activity or deoxyglucose uptake) is retained in the SCN of diurnal species (*Sato and Kawamura, 1984*; *Schwartz, 1991*; *Ruby and Heller, 1996*). This has led to the view that the basic mechanisms involved in circadian timekeeping in the SCN are conserved between species with different activity patterns (diurnal or nocturnal). By extension, this implies that neural mechanisms underlying temporal niche preference may be located downstream from the SCN (*Smale et al., 2003*; *Smale et al., 2008*). However, the information provided by extracellular recordings is limited, revealing only the daily variation in SCN neuronal population activity, but offering no understanding of the electrophysiological mechanisms involved or the electrical properties of single neurons, particularly on how they respond to inputs. The extent to which SCN neurophysiology in diurnal species is altered to adapt neuronal function to the specific demands of the animal's temporal niche preference remains unknown. For example, day-active animals are exposed to daytime light (the main excitatory input to the SCN) to an extent that nocturnal species are not (*Yan et al., 2020*). On the other hand, behavioral feedback to the clock from arousal and wakefulness occurs at different times of the circadian day depending on the temporal niche that an animal occupies (at night and during the day for nocturnal and diurnal animals, respectively) (*Hughes and Piggins, 2012*; *Jha et al., 2021*). To date, there have been no whole-cell recordings of SCN neurons from a diurnal species, and the question of how, or if, SCN neurophysiology is altered to accommodate a diurnal lifestyle remains unanswered. *Rhabdomys pumilio* (the four striped mouse) represents an excellent opportunity to address this question. This species is primarily day-active (*Dewsbury and Dawson, 1979*; *Schumann et al., 2005*; *Mallarino et al., 2018*; *Bano-Otalora et al., 2021*), and is a murid rodent, facilitating comparison with established findings from closely related nocturnal species (mice and rats). The designation of *R. pumilio* as a reliably day-active species is supported by other aspects of its biology, particularly its visual system which has several adaptations (cone rich retina, UV blocking lens) that are typical of animals relying on daytime vision (*Allen et al., 2020*).

We adopted a parallel approach of experimental recording and advanced computational modeling to understand the *R. pumilio* SCN. First, we address the lack of data on single-cell physiology in diurnal SCN by using whole-cell recordings to describe spontaneous electrical states and their daily variation. We then determined the evoked membrane properties of these diurnal SCN neurons by recording their responses to inputs. We then turned to cutting-edge data assimilation and modeling approaches to gain insight into the cellular and ionic mechanisms underlying passive and evoked electrical states. Our results revealed similarities in SCN neurophysiology between the *R. pumilio* and other rodent species, but also exposed fundamental differences which may serve to accommodate SCN functioning to a diurnal niche.

## Results

### SCN neuropeptidergic organization in the diurnal *Rhabdomys pumilio*

Prior to assaying single-cell electrical properties in the *R. pumilio* SCN, we first described the anatomical and neuropeptidergic organization of the SCN in this species. This provided us with a practical guide to ensure only neurons within the SCN were targeted for electrophysiology since no brain atlas yet exists for this species. To this end, we performed immunofluorescence labeling for nuclear DNA with DAPI, and vasoactive intestinal polypeptide (VIP), arginine vasopressin (AVP), and gastrin-releasing peptide (GRP) (*Figure 1*), which are the main neuropeptides expressed in the SCN, delineate its anatomical boundary, and are critical for circadian rhythm generation and coordination (*Abrahamson and Moore, 2001*; *Ono et al., 2021*).

The gross neuroanatomy of the *R. pumilio* SCN across the rostro-caudal axis is broadly similar to other rodent species (*Smale and Boverhof, 1999*; *Abrahamson and Moore, 2001*; *Figure 1A*). Immunofluorescence labeling for the main neuropeptides showed that the *R. pumilio* SCN contains VIP, AVP, and GRP, and importantly, the neuroanatomical localization of these neuropeptides was broadly similar to the distribution found in other rodent species (*Smale and Boverhof, 1999*; *Abrahamson and Moore, 2001*), AVP-positive cell bodies were mainly localized in the dorsomedial aspect (sometimes termed 'shell' (*Figure 1B*)), while VIP-positive somas were localized throughout

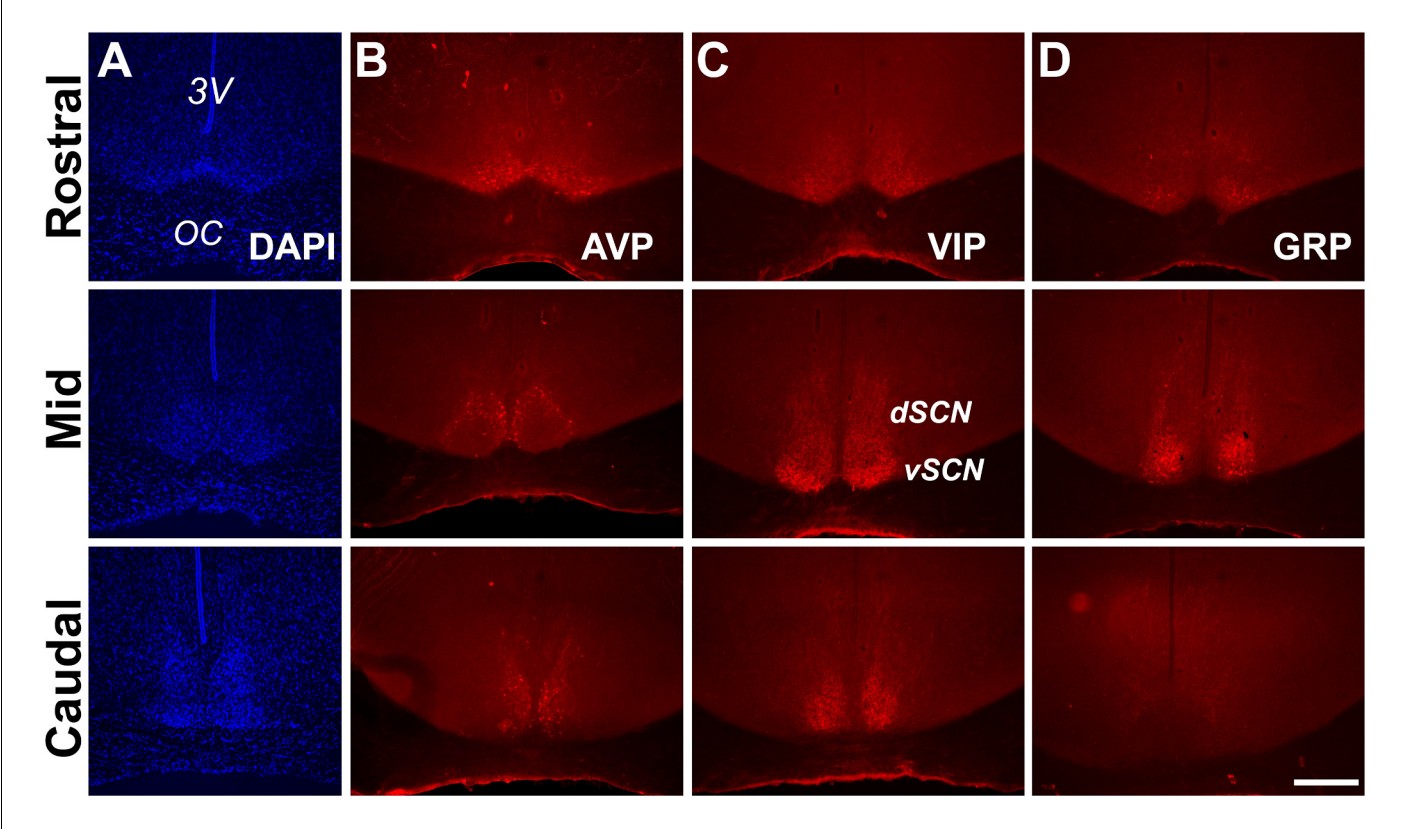

**Figure 1.** Anatomy and neuropeptidergic organization of the *Rhabdomys pumilio* SCN. (A) Coronal sections of the *R. pumilio* SCN taken across the rostro-caudal axis labeled with DAPI, and immunofluorescence for the main SCN neuropeptides: (B) Arginine-vasopressin (AVP), (C) Vasoactive intestinal peptide (VIP) and (D) Gastrin releasing peptide (GRP). 3V: third ventricle; OC: optic chiasm. dSCN: dorsal SCN, vSCN: ventral SCN. Labeling at the rostral level applies to mid and caudal aspects. Scale bar: 250 μm.

the ventral region or 'core', with VIP immunoreactive axonal processes extended into the dorsal SCN (*Figure 1C*). By contrast, GRP-positive neurons were localized in the central SCN (*Figure 1D*).

## Diurnal changes in the spontaneous electrical activity of *Rhabdomys pumilio* SCN neurons

The day-night electrical activity and membrane excitability states of SCN neurons at the single-cell level are well characterized in nocturnal animals (*Colwell, 2011*; *Belle and Diekman, 2018*; *Harvey et al., 2020*), but thus far there are no such measurements performed in the SCN of diurnal mammals. We therefore set out to describe the intrinsic electrical states of *R. pumilio* SCN neurons with respect to the cell's passive membrane properties (resting membrane potential [RMP], spontaneous firing rate [SFR], and input or membrane resistance [$R_{input}$]), and how these change across the day and at night, using in vitro whole-cell patch clamp electrophysiology.

Recording (*Figure 2A*) from a total of 111 SCN neurons (from eight animals) over the day-night cycle revealed several spontaneous excitability states in *R. pumilio* (*Figure 2B*), similar to previous descriptions in mice (*Belle et al., 2009*; *Diekman et al., 2013*; *Paul et al., 2016*; *Collins et al., 2020*). Thus, some SCN neurons were resting at moderate RMPs (−43.9 ±0.41 mV, n=94/111) and firing action potentials (APs). Other neurons were severely depolarized or 'hyperexcited' (−32.7 ± 2.36, n=6/111), to the extent that rather than generating APs, they became depolarized-silent or exhibited depolarized low-amplitude membrane oscillations (DLAMOs). The final category of neurons were hyperpolarized-silent, having RMPs too negative to sustain firing (−50.5 ± 2.29 mV, n=11/111).

SCN neurons were overall more excited during the day than at night (*Figure 2C–E*), with hyperpolarized-silent neurons appearing at night, and the daytime state being characterized by firing and

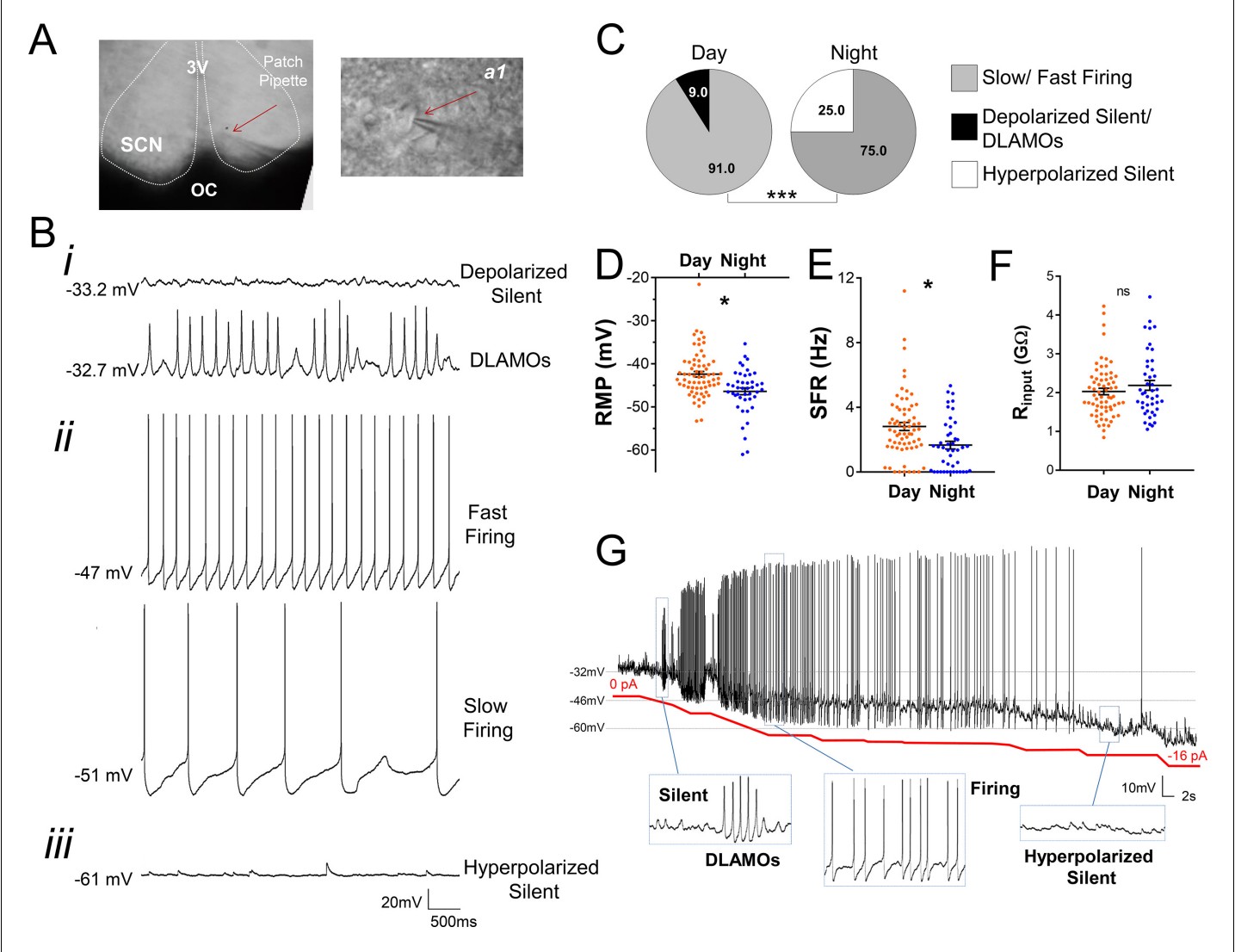

**Figure 2.** Diurnal changes in the spontaneous electrical activity of *Rhabdomys pumilio* SCN neurons. (A) Whole-cell patch clamp recording setup showing bright-field image of a SCN coronal brain slice. The SCN (delineated by white dotted lines) can be observed above the optic chiasm (OC), on either side of the third ventricle (3V). Patch pipette targeting a SCN neuron is indicated by the red arrow and magnified in inset (a1). (B) Representative current-clamp traces of the different spontaneous excitability states recorded in *R. pumilio* SCN neurons: (i) highly depolarized cells, becoming silent (top trace) or displaying depolarized low-amplitude membrane oscillations (DLAMOs) (bottom trace); (ii) moderate resting membrane potential (RMP) with cells firing action potentials (APs) at high or low rate; and (iii) hyperpolarized-silent neurons. (C) Pie charts showing the percentages of SCN neurons in the different electrical states during the day and at night ($\chi^2$=21.498, ***p<0.001, Chi-square test). Mean RMP (D), spontaneous firing rate (SFR) (E) and input resistance ($R_{input}$) (F) of neurons recorded during the day (orange, n=67 for RMP and SFR, n=66 for $R_{input}$) and at night (blue, n=44 for RMP and SFR, n=43 for $R_{input}$). Data are expressed as mean ± SEM with each dot representing an individual neuron. *p < 0.05, ns: non-significant. RMP: $F_{(1, 5.036)}$=10.249, p=0.024; SFR: $F_{(1, 7.027)}$=5.998, p=0.044; $R_{input}$: $F_{(1, 5.984)}$=0.878, p=0.385, mixed-effects linear model. (G) Manual hyperpolarization of hyperexcited SCN neuron elicits a range of electrical states. Silent cell resting at highly depolarized state could be driven to display DLAMOs, fire APs, and become hyperpolarized-silent by injection of progressive steps of steady-state hyperpolarizing currents (from 0 to ~ −16 pA (red line); driving RMP from −32 mV to −60 mV).

The online version of this article includes the following source data for figure 2:

**Source data 1.** Numerical data to support graphs in *Figure 2C–F*.

depolarized cells, indicating a time-of-day control on these cellular electrical states ($\chi^2$=21.498, p<0.001, Chi-square test, *Figure 2C*), as reported in the mouse SCN (*Belle et al., 2009*; *Diekman et al., 2013*; *Paul et al., 2016*; *Belle and Piggins, 2017*). Accordingly, RMP and SFR showed a robust circadian variation (*Figure 2D–E*). During the day, SCN neurons were overall

resting at more depolarized RMP, generating APs at a higher rate. This indicates that, as in nocturnal species (*Belle et al., 2009*; *Belle and Piggins, 2017*), cellular RMP in the diurnal *R. pumilio* SCN is a strong determinant of electrical states and SFR. To directly test this, we subjected depolarized-silent SCN neurons (n=2, resting at ~−30 mV) to progressive steps of steady-state suppressive (negative) currents, to see if we could elicit the range of spontaneous electrical behaviors seen in SCN neurons. Indeed, *R. pumilio* SCN neurons could be easily driven to transit from the depolarized- through to hyperpolarized-silent states, switching to DLAMOs and firing activity at appropriate RMPs in the process (*Figure 2G*).

Measurement of $R_{input}$ values showed a range from 0.84 to 4.47 GΩ, skewed toward high values, as reported in other species (*Pennartz et al., 1998*; *Jackson et al., 2004*; *Kuhlman and McMahon, 2004*; *Belle et al., 2009*). However, we found neither a significant day-night variation in this measure ($F_{(1, 5.984)}$=0.878 p=0.385, *Figure 2F*) nor a correlation with RMP ($R^2 = 0.0305$, p=0.0695, n=109),

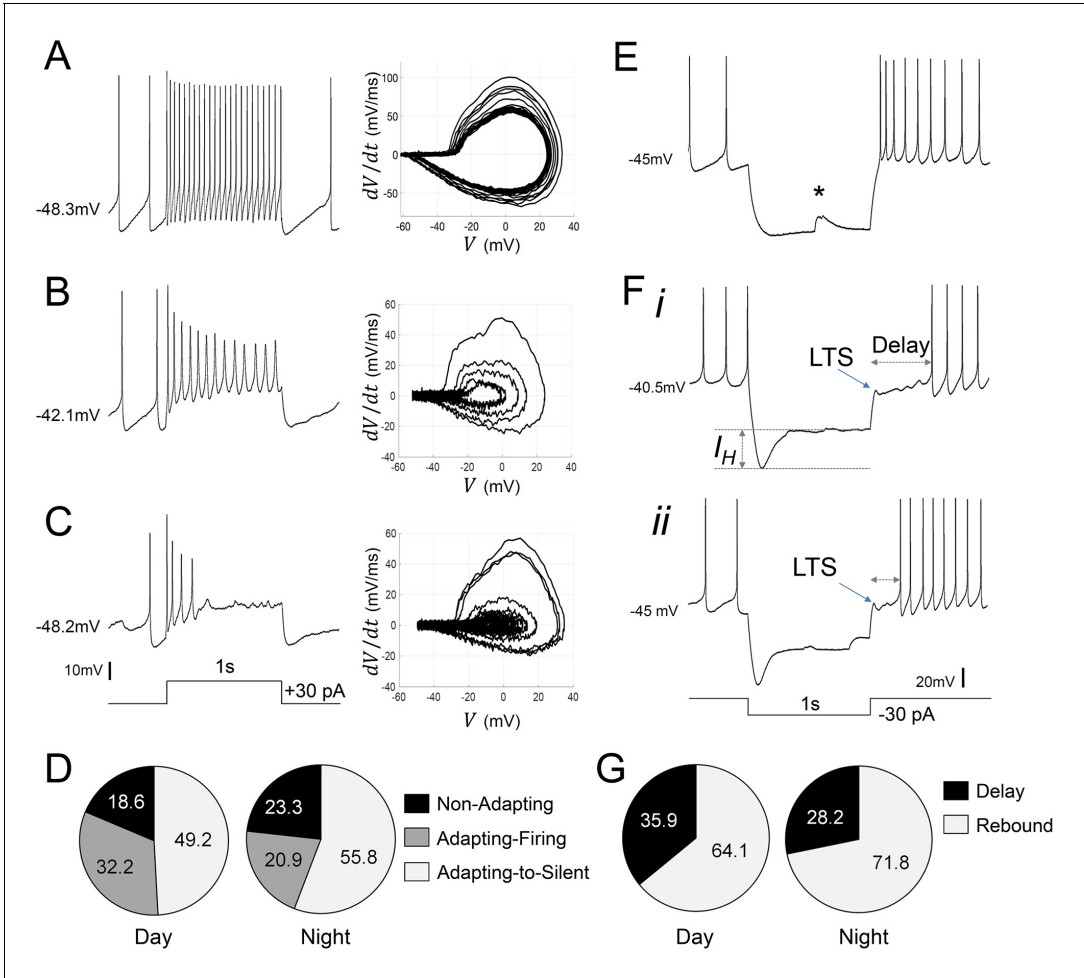

**Figure 3.** Diverse responses to depolarizing and hyperpolarizing current pulses in *Rhabdomys pumilio* SCN neurons. Representative current-clamp traces showing the different type of responses to a depolarizing pulse (1 s, +30 pA): (A) non-adapting; (B) adapting-firing; or (C) adapting-to-silent response. Phase–plot diagrams on the right of each panel (A, B, or C) show action potential (AP) velocity, trajectory and rate of frequency adaptation during the pulse for these neurons. (D) Pie charts showing the percentage of recorded neurons displaying each of these responses to depolarizing pulses during the day and at night (χ2=1.621, *p*=0.4447). (E–F) Representative current-clamp traces showing the different type of responses to a 1 s, −30 pA hyperpolarizing pulse: (E) Type-A cells responded with a rebound spike upon termination of the pulse; (F) Type-B cells exhibited a rebound hyperpolarization which produced a delay-to-fire, following a LTS ((i–ii) long and short delay, respectively). (G) Pie charts showing the percentage of cells displaying a rebound spike or a delay-to-fire response during the day and at night (χ2=0.6552, p=0.4183). * indicates a spontaneous synaptic input. LTS: low threshold spike. $I_H$: inward membrane rectification or depolarizing 'sag'.

The online version of this article includes the following source data for figure 3:

**Source data 1.** Numerical data to support graphs in *Figure 3D and G*.

which stands in contrast to measurements in the SCN of nocturnal animals (*de Jeu et al., 1998*; *Kuhlman and McMahon, 2004*; *Belle et al., 2009*). This represents the first substantial difference between *R. pumilio* and mouse or rat SCN.

## Diversity in the evoked electrical responses of *Rhabdomys pumilio* SCN neurons

In addition to the daily variation in intrinsic electrical activity, SCN clock function also critically relies on the integrated activity of excitatory and inhibitory synaptic signals (*Albers et al., 2017*). These inputs originate both from within the SCN (e.g. excitation or inhibition via GABA-GABA$_A$ receptor signaling) and from other brain circuits (e.g. excitation or inhibition via glutamate, or GABA signaling). Mimicking these fast signals by depolarizing and hyperpolarizing current pulses elicits diverse electrical responses in the SCN of nocturnal animals and is useful for characterizing SCN neurons (*Pennartz et al., 1998*; *Belle et al., 2009*; *Harvey et al., 2020*). Therefore, we next investigated the spiking responses of *R. pumili*o SCN neurons to inputs by challenging the cells with brief current pulses (see Materials and methods).

When subjected to depolarizing pulses, *R. pumilio* SCN neurons exhibited electrical responses similar to those of nocturnal species: a small proportion of cells (21/102) responded with a sustained and regular train of action potentials, with no, or marginal, spike-frequency adaptation (non-adapting cells, *Figure 3A*). The remaining neurons (81/102) showed some degree of frequency adaptation (*Figure 3B,C*). These cells either progressively slowed firing rate and exhibited increased spike shape broadening and amplitude reduction during the pulse (adapting-firing, *Figure 3B*), or fired only a few APs during the initial phase of the depolarization before entering a silent state (adapting-to-silent, *Figure 3C*). We found non-adapting and adapting cells resting at similar RMPs, indicating that cellular RMP was not the determinant of response type (e.g. *Figure 3A* vs C). The proportion of cells displaying each of these responses did not vary across the day-night cycle ($\chi^2$=1.621, p=0.4447, Chi-square test, *Figure 3D*). This suggests that, as in the mouse SCN (*Belle et al., 2009*; *Belle and Piggins, 2017*), these different types of spiking behavior likely reflect 'hardwired' differences between SCN neurons, rather than time-of-day-dependent variations in physiological state.

We next mimicked the effect of inhibitory signals by injecting hyperpolarizing current pulses (*Figure 3E,F*). In all cases, spike firing ceased during these hyperpolarizing currents. Upon pulse termination, 67% (69/103) of *R. pumilio* SCN neurons immediately resumed normal firing or showed rebound depolarization spiking before resuming normal pre-pulse level of firing (*Figure 3E*), as previously reported for mouse and rat SCN (*Thomson and West, 1990*; *Pennartz et al., 1998*; *Kuhlman and McMahon, 2004*; *Belle et al., 2009*). The remaining 33% (34/103) of units displayed a low-threshold spike (LTS) followed by a rebound hyperpolarization which produced a prominent delay, ranging from 160 to 1430 ms, before firing resumed (*Figures 3Fi–ii* and 6H). A high proportion of cells in this second group (73.5%; 25/34) also showed an inward rectification or depolarization 'sag' (*Figure 3F*) during the pulse, an electrical response that is associated with H-current activation ($I_H$, [*Pennartz et al., 1998*; *Atkinson et al., 2011*]). The hyperpolarization-evoked delay to fire and LTS response (*Figure 3Fi–ii*) have not previously been reported for SCN neurons, and thus represents another significant point of divergence in SCN neurophysiology between *R. pumilio* and, previously studied, nocturnal species.

We termed *R. pumilio* neurons with rebound firing Type-A cells (*Figure 3E*), and those with delays Type-B neurons (*Figure 3Fi–ii*), to be consistent with nomenclatures previously used to identify neurons with those distinct electrical characteristics elsewhere in the brain (*Burdakov and Ashcroft, 2002*; *Burdakov et al., 2004*). The relative abundance of Type-A and -B cells did not change across the day-night cycle ($\chi$2=0.6552, p=0.4183, Chi-square test, *Figure 3G*), indicating that these response properties are determined by cell-type rather than time-of-day.

In addition, analysis of the relationship between cellular responses to depolarizing and hyperpolarizing pulses revealed that non-adapting and adapting neurons exhibited similar proportions of rebound or delay-to-fire behaviors (~34–38% delay and ~62–66% rebound), suggesting that non-adapting or adapting responses in these neurons do not determine firing characteristics to hyperpolarizing pulses.

### Ionic mechanisms underlying evoked electrical responses

A comprehensive understanding of SCN neurophysiology would encompass an appreciation of the ionic mechanisms and channel parameters responsible for the electrophysiological properties revealed in our whole-cell recordings (*Belle and Diekman, 2018*; *Harvey et al., 2020*). Capturing this ionic information from current-clamp data has only recently become feasible due to advances in data assimilation (DA) techniques (*Abarbanel, 2013*). Here, we developed a state-of-the-art DA algorithm (see Materials and methods section and Appendix 1 for a detailed description) and applied it to build detailed computational models of *R. pumilio* SCN neurons (*Figure 4* and *Figure 4—figure supplements 1–2*). This modeling approach reproduced the voltage trajectory and nuances of action potentials and subthreshold electrical activity generated during spontaneous and evoked firing of SCN neurons in remarkable detail (*Figure 4* and *Figure 4—figure supplement 2*),

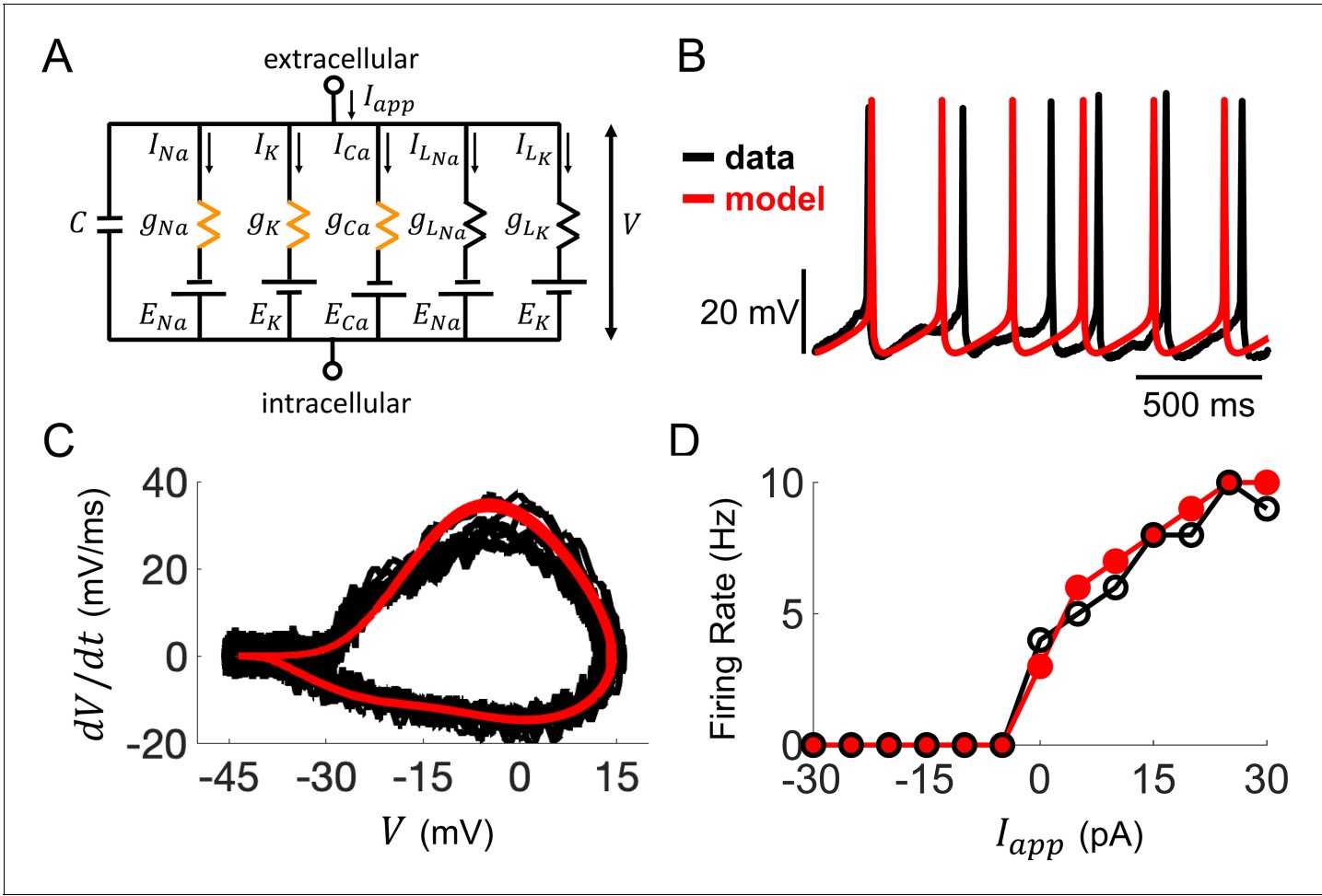

**Figure 4.** Computational modeling of *Rhabdomys pumilio* SCN neurons. (**A**) Schematic of conductance-based model for *R. pumilio* SCN neurons containing sodium ($I_{Na}$), calcium ($I_{Ca}$), potassium ($I_K$), and leak ($I_{LNa}$, $I_{LK}$) currents. Orange resistors ($g_{Na}$, $g_{Ca}$, $g_K$) indicate voltage-gated conductances, black resistors ($g_{LK}$, $g_{LNa}$) indicate passive leak conductances. (**B**) Voltage traces showing similarity in spontaneous firing of action potentials (APs) in the model (red) compared to a current-clamp recording from a *R. pumilio* SCN neuron (black). (**C**) Phase-plot of the derivative of voltage with respect to time (*dV/dt*) as a function of voltage (*V*) depicting the shape of APs in the model (red) and the current-clamp recording (black) during spontaneous firing. (**D**) Similarity in firing rate of the model (red) and current-clamp recordings (black) as a function of applied current ($I_{app}$).

The online version of this article includes the following figure supplement(s) for figure 4:

**Figure supplement 1.** Example current-clamp traces used in data assimilation algorithm for building computational models of *Rhabdomys pumilio* SCN neurons.

**Figure supplement 2.** Example voltage traces for a computational model of *Rhabdomys pumilio* SCN neurons fit using a data assimilation algorithm.

**Figure supplement 3.** Ionic currents underlying action potential generation in computational models of *Rhabdomys pumilio* and mouse SCN neurons.

**Figure supplement 4.** Bifurcation diagram for a computational model of *Rhabdomys pumilio* SCN neurons.

providing confidence that the ionic currents and parameters estimated by our DA algorithm, and their dynamical relationship in the models, are indeed a close match to their biological values and activity.

Through simulations of the model, we first assessed how ionic conductances interact with each other to produce AP firing and other electrical behaviors. We applied this approach to compare the conductances underlying spontaneous AP generation in the *R. pumilio* SCN model (*Figure 4A*) to our previously published model of mouse SCN neurons (*Belle et al., 2009*) containing the same sets of ionic currents (voltage-dependent transient sodium $I_{Na}$, voltage-dependent transient calcium $I_{Ca}$, voltage-dependent potassium $I_K$, and voltage-independent leak, $I_L$). We found that the overall profile of how these currents contribute to AP generation is similar across the two species (*Figure 4—figure supplement 3*). In addition, the types of bifurcations at the transitions between rest states and spiking are the same in both models (subcritical Hopf from hyperpolarized silent to spiking, and supercritical Hopf from depolarized-silent to spiking), suggesting the qualitative dynamics that lead to repetitive AP firing are similar across the two species (*Figure 4—figure supplement 4A*). Furthermore, the *R. pumilio* model can produce all the electrical behaviors observed across the day-night cycle (depolarized-silent, DLAMOs, fast-firing, slow-firing, and hyperpolarized-silent, *Figure 4—figure supplement 4B–F*) through an antiphase circadian rhythm in sodium and potassium leak currents (with higher sodium leak current during the day than at night), consistent with the 'bicycle model' proposed for the circadian regulation of electrical activity in mice and flies (*Flourakis et al., 2015*).

We next used the model to gain insight into the mechanisms responsible for the adapting versus non-adapting firing behaviors observed in response to depolarizing pulses. Our DA algorithm yielded models that faithfully reproduced the voltage traces and spike shapes from non-adapting, adapting-firing, and adapting-to-silent cells (*Figure 5A–B*). By inspecting the ionic currents flowing during the simulated voltage traces, we assessed the role of voltage-gated sodium $I_{Na}$, calcium $I_{Ca}$, and potassium $I_K$ currents in producing these responses (*Figure 5C*).

Our models revealed that frequency adaptation in SCN neurons in response to excitation resulted from the progressive inactivation of sodium channels. Indeed, the adapting-firing model indicated a much smaller amount of $I_{Na}$ available for the APs during the depolarizing pulse (peak $I_{Na}$ = -80pA, *Figure 5Cii*), and a greater reduction in sodium conductance $G_{Na}$ (26 nS before vs 1.5 nS during the pulse, *Figure 5Dii*) compared with the non-adapting model (peak $I_{Na}$ = −580 pA; $G_{Na}$ = 27 nS before vs 13 nS during the pulse, *Figure 5Ci,Di*). Remarkably, however, increased sodium channel inactivation ($h_{Na}$ close to 0) could not be ascribed to intrinsic differences in the sodium channel properties themselves between the non-adapting and adapting-firing models as the kinetic parameters of the sodium activation and inactivation gating variables were similar (*Figure 5F–G*). Rather, the difference was due to differing properties of the potassium channels. A combination of a flattened steady-state potassium activation ($n_\infty$) curve (*Figure 5H*) and the lower $g_K$ value (*Figure 5I*), led to a smaller $I_K$ and reduced $G_K$ during AP firing in the adapting-firing compared to the non-adapting model (250 pA, 3 nS vs 900 pA, 11 ns, respectively) (*Figure 5C-D i-ii*). Since $I_K$ is an outward current, this means that the adapting-firing model does not repolarize as strongly after the peak of an AP, and therefore, the membrane does not hyperpolarize enough to de-inactivate the sodium channels. Thus, in the adapting-firing model, the inability of a weak $I_K$ to sufficiently repolarize the membrane is what ultimately leads to the reduced $I_{Na}$ and low-amplitude APs. The $I_K$ is even smaller in the adapting-to-silent model (*Figure 5Ciii*), failing to repolarize the membrane, and leads to sustained inactivation of the sodium channel (*Figure 5Eiii*), negligible sodium conductance (*Figure 5Diii*) and ultimately the inability to repeatedly fire APs during the pulse (*Figure 5Biii*). In summary, our models support progressive sodium channel inactivation as the mechanism of frequency adaptation (consistent with experimental observation in neurons elsewhere in the brain [*Fleidervish et al., 1996*; *Jung et al., 1997*; *Kimm et al., 2015*] and our previously published model of mouse SCN neurons [*Belle et al., 2009*]), while indicating that this is primarily a consequence of a weak $I_K$.

We next interrogated our models for the key ionic origins of Type-A vs Type-B responses to inhibition (*Figure 3E,F*). In both cell types, hyperpolarizing pulses drove the membrane potential in the real and model cells below the firing threshold, which suppressed firing activity during the pulse (*Figure 6A–B,i–ii*). Model analysis showed that in the Type-A cell, the $I_{Na}$ and $I_{Ca}$ currents were larger during the first AP immediately following the pulse than during the APs before the pulse (*Figure 6D*), leading to a high-amplitude rebound spike. The rebound spiking was due to sodium and calcium ion channels becoming completely de-inactivated ($h_{Na}$ and $h_{Ca}$ both approach 1) at the

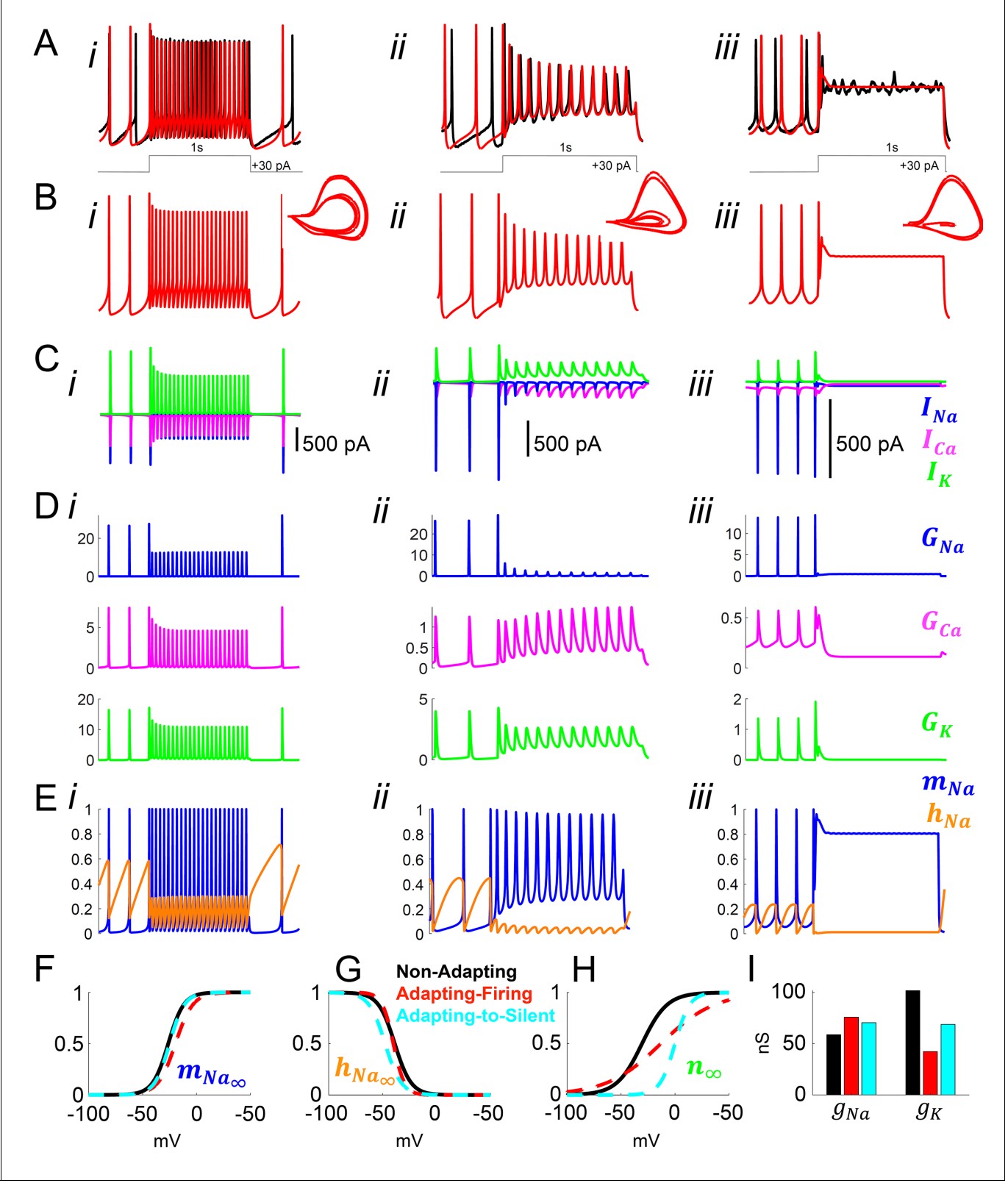

**Figure 5.** Model simulation of the responses to depolarizing pulses in *Rhabdomys pumilio* SCN neurons and the underlying ionic mechanisms. (A–B) Voltage traces of models (red) and current-clamp recordings (black) during depolarizing pulses (1 s, +30 pA) showing non-adapting (i), adapting-firing (ii), and adapting-to-silent (iii) responses. (C) Ionic currents sodium ($I_{Na}$, blue), calcium ($I_{Ca}$, magenta), and potassium ($I_K$, green) in the models during the non-adapting (i), adapting-firing (ii), and adapting-silent (iii) responses. (D) Ionic conductances for sodium ($G_{Na}$, blue), calcium ($G_{Ca}$, magenta), and

*Figure 5 continued on next page*

Figure 5 continued

potassium ($G_K$, green) in the models during the non-adapting (i), adapting-firing (ii), and adapting-silent (iii) responses. (E) Sodium activation ($m_{Na}$, blue) and inactivation ($h_{Na}$, orange) gating variables in the models during the non-adapting (i), adapting-firing (ii), and adapting-silent (iii) responses. Ions cannot pass through the channel if it is closed ($m_{Na}$ = 0) or inactivated ($h_{Na}$ = 0); maximal current flows when the channel is fully open ($m_{Na}$ = 1) and fully de-inactivated ($h_{Na}$ = 1). Steady-state gating variables as a function of voltage in the non-adapting (black), adapting-firing (red), and adapting-to-silent (cyan) models for (F) sodium activation ($m_{Na\infty}$), (G) sodium inactivation ($h_{Na\infty}$), and (H) potassium activation ($n_\infty$). The flattening of the $n_\infty$ curve in the adapting-firing model indicates that the channel is less activated at depolarized voltages than the non-adapting model (e.g. at −13 mV, the adapting-firing model is only half activated ($n_\infty$ = 0.5), whereas the non-adapting model is almost fully activated ($n_\infty$ = 0.93)). (I) Maximal conductance parameters $g_{Na}$ and $g_K$ in the non-adapting (black), adapting-firing (red), and adapting-to-silent (cyan) models. Notice that the maximal potassium conductance parameter is much smaller in the adapting-firing model ($g_K$ = 43 nS) than in the non-adapting model ($g_K$ = 102 nS).

hyperpolarized membrane potential reached during the pulse (*Figure 6F*). The time scale of calcium ion channel inactivation causes $I_{Ca}$ to remain elevated for a few hundred milliseconds after the pulse, resulting in a transient after-depolarization and a short burst of firing before returning to the baseline pre-pulsed spike rate (*Figure 6A*), consistent with experimental observation elsewhere in the hypothalamus (*Burdakov and Ashcroft, 2002*).

Similar $I_{Na}$ and $I_{Ca}$ dynamics were present in the Type-B neuron model. However, the rebound hyperpolarization and prominent delay-to-fire after the pulse observed in Type-B neurons (*Figure 3F* and *6Bi-ii*), was not possible to reproduce using our existing basic model (*Figure 4A*), consistent with the failure to observe such behavior in the mouse SCN. It is well established in neurons elsewhere in the brain that the inhibitory actions of the transient subthreshold activating A-type ($I_A$) voltage-gated potassium channels (Kv) underpin such delay-to-fire activity (*Schoppa and Westbrook, 1999*; *Saito and Isa, 2000*; *Burdakov and Ashcroft, 2002*; *Burdakov et al., 2004*; *Nadin and Pfaffinger, 2010*; *Tarfa et al., 2017*). Another feature of Type-B activity that could not be recreated with our basic model was the prominent depolarization 'sag' seen in the voltage trace during the pulse (*Figure 3F* and *6Bi*). Such behavior could be produced by activation of an $I_H$ current by the hyperpolarizing pulse. We therefore added $I_A$, as well as a hyperpolarization-activated ($I_H$) current, to our mouse SCN model in an attempt to recreate the voltage trace and biophysical condition of the Type-B neuron (*Figure 6C*).

The expanded model revealed a larger $I_A$ current during the first APs after the delay (480 pA) than during a typical spike (220 pA, *Figure 6E*). Importantly, there was also 15 pA of $I_A$ current flowing during the delay itself (*Figure 6E* inset). It is noteworthy that this was greater than the 5 pA of $I_A$ current that flows during the inter-spike interval. This enhanced $I_A$ current following the pulse was due to de-inactivation of the A-type channel ($h_A$ approaches 1) during the hyperpolarizing pulse (*Figure 6G*), rendering the $I_A$ channel fully available upon release of the pulse, an observation that is consistent with experimental findings (*Burdakov et al., 2004*). The $I_A$ current then inactivates slowly and, until this outward current decays sufficiently, the cell cannot reach threshold to fire, thereby prolonging inhibition. This inhibition-supportive action of $I_A$ is consistent with observations made elsewhere in the brain (*Burdakov and Ashcroft, 2002*; *Burdakov et al., 2004*; *Tarfa et al., 2017*), and previous simulations (*McCormick and Huguenard, 1992*; *Rush and Rinzel, 1995*; *Patel et al., 2012*). To confirm our model's prediction that $I_A$ current is indeed responsible for the delay in *R. pumilio* SCN neurons, we used pharmacological blockade of $I_A$ channels with bath application of 4-Aminopyridine (4-AP) (*Figure 6H–I*). As predicted, 5 mM 4-AP reversibly eliminated the rebound hyperpolarization and prominent delay-to-fire in all Type B neurons tested, irrespective of the delay duration ($F_{(2, 10)}$ = 14.09, p=0.0012, one way RM-ANOVA, n=6), reducing the average latency to the first spike after a 1 s −30 pA pulse from 0.596 ± 0.134 s to 0.086 ± 0.013 s (Baseline vs 4-AP, p=0.0024 Tukey's multiple comparisons test).

It has previously been shown that variation in cellular $I_A$ conductances and inactivation time constant can impact time to fire (e.g. *Saito and Isa, 2000*; *Tarfa et al., 2017*), and this may explain the broad range in the delay-to-fire, from 160 to 1430 ms, seen in our Type-B neurons (*Figure 6J*). Indeed, this was the case in our model. By varying the maximal $I_A$ conductance (*Figure 6M,O*) and inactivation time constant (*Figure 6N,P*) parameters, we were able to capture the full range of latency to fire seen in Type-B cells, with higher conductances and longer inactivation time constants producing longer delays. Consistent with our experiments, complete removal of the $I_A$ conductance

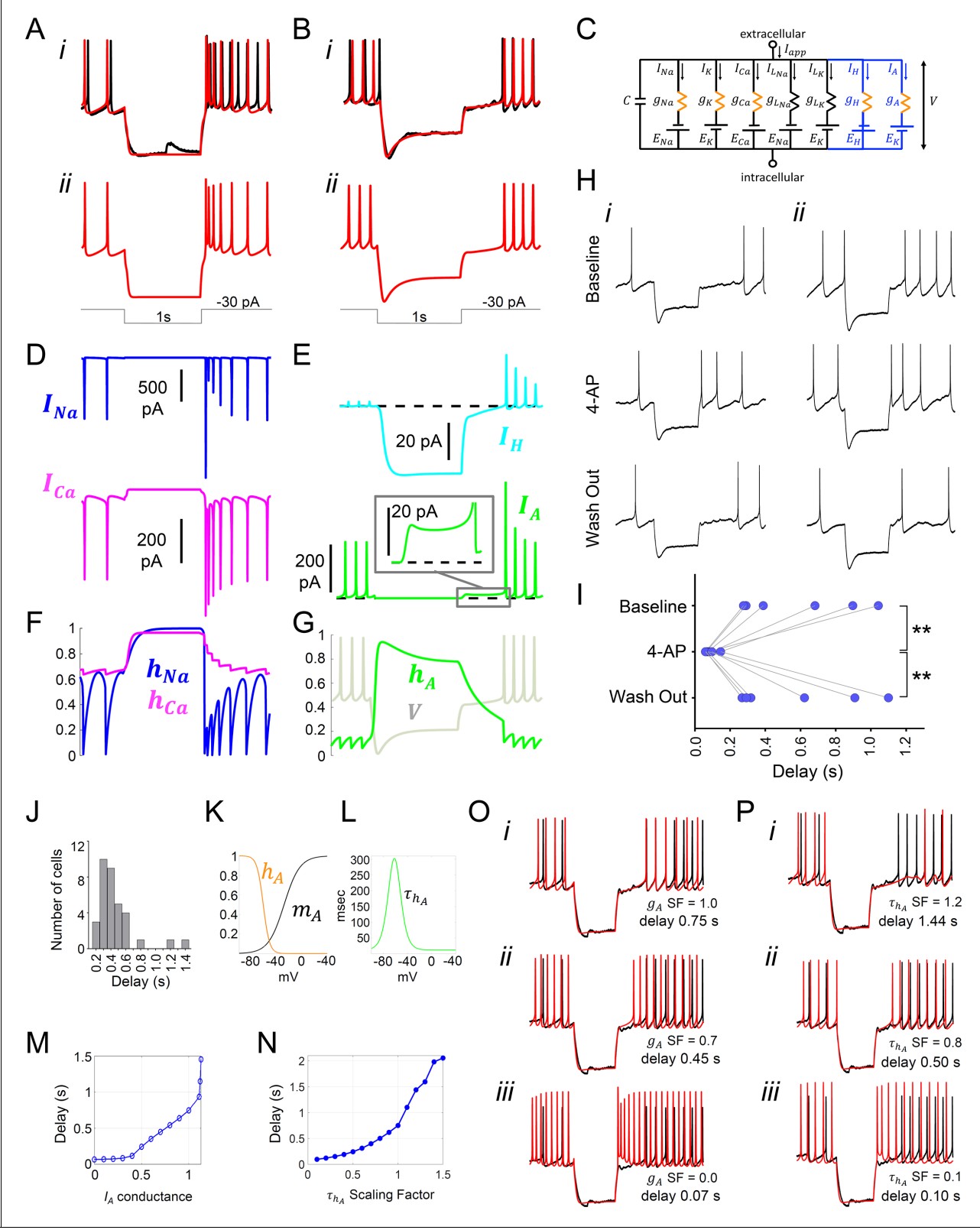

**Figure 6.** Model simulation of the responses to hyperpolarizing pulses in *Rhabdomys pumilio* SCN neurons and the underlying ionic mechanisms. (A–B) Voltage traces of models (red) and current-clamp recordings (black) during hyperpolarizing pulses (1s, -30 pA) showing rebound spiking of Type-A neurons (A) and delay responses of Type-B cells (B). (C) Schematic of conductance-based model for Type-B *R. pumilio* SCN neurons showing the addition of transient potassium ($I_A$) and hyperpolarization-activated ($I_H$) currents (blue). (D) Ionic currents for sodium ($I_{Na}$, blue) and calcium ($I_{Ca}$,

*Figure 6 continued on next page*

*Figure 6 continued*

magenta) in the model during the Type-A neuronal rebound spiking response. (E) Ionic currents $I_H$ (cyan) and $I_A$ (green) in the model during the delay response of Type-B neurons. (F) Sodium ($h_{Na}$, blue) and calcium ($h_{Ca}$, magenta) inactivation gating variables in the model during the Type-A neuronal rebound spiking response. (G) Transient potassium ($I_A$) inactivation gating variable ($h_A$, green) in the model during the delay response in Type-B neurons (voltage trace, $V$, is indicated in gray and is the same V-trace shown in B). (H) Representative current-clamp recordings of *R.pumilio* Type-B neurons during hyperpolarizing pulses (1s, -30 pA) showing a long (i) and short (ii) delay-to-fire latency under baseline conditions (top trace). Bath application of 5mM 4-Aminopyridine (4-AP, A-type channel blocker) eliminated the rebound hyperpolarization and prominent delay-to-fire, so neurons exhibited a rebound spike upon termination of the pulse (mid traces). Delay-to-fire behavior returned after blocker washout (bottom traces). (I) Summary plot of the latencies to fire upon termination of the pulse in Type-B cells (n=6, 2 animals) under baseline conditions, in the presence of 4-AP, and during washout. **p<0.01, One-way repeated measures ANOVA followed by Tukey's post hoc test. (J) Histogram showing delay-to-fire latencies measured in Type-B cells. (K–L) Gating variable functions for model $I_A$ current: (K) steady-state activation ($m_A$, black), steady-state inactivation ($h_A$, orange), and (L) inactivation time constant ($\tau_{h_A}$, green). (M) Relationship between $I_A$ conductance ($g_A$ Scaling Factor) and delay-to-fire latencies in model of Type-B cells. (N) Relationship between the time constant of $I_A$ inactivation and delay-to-fire latencies in model of Type-B cells. (O) Data trace for a cell with a 0.75 s delay (black) overlaid with model voltage traces (red) with varied amounts of $I_A$ conductance: (i) model of Type-B cell with $g_A$ SF = 1 exhibiting a 0.75 s delay; (ii) Model from (i) with reduced $I_A$ conductance ($g_A$ SF = 0.7) exhibiting a reduced delay-to-fire latency; (iii) Model from (i) with no $I_A$ current ($g_A$ SF = 0), exhibiting rebound spiking, as in Type-A neurons. $g_A$ SF: $g_A$ Scaling Factor. (P) Model simulations for $I_A$ inactivation time constant scaling factors of 1.2 (i), 0.8 (ii) and 0.1 (iii). $\tau_{h_A}$ SF: $\tau_{h_A}$ Scaling Factor.

The online version of this article includes the following source data for figure 6:

**Source data 1.** Numerical data to support graphs in *Figure 6I*.

eliminated the delay and produced a Type-A response (*Figure 6Oi–iii*), reinforcing the different ionic composition of these two cell types.

In summary, our revised model was able to mirror all the electrical features observed in *R. pumilio* SCN neurons in response to extrinsic inputs, and identified transient subthreshold A-type potassium channels as playing a key role in evoked-suppression firing in simulated SCN neurons, which we subsequently confirmed with pharmacology.

## $I_A$ currents suppress firing under physiological simulation

We finally interrogated our model to understand how the $I_A$ conductances required to explain SCN responses to hyperpolarizing pulses may impact firing activity in a more realistic neurophysiological setting. To this end, we first subjected the model to simulated synaptic conductances recorded from *R. pumilio* SCN neurons (*Figure 7* and *Figure 7—figure supplement 1*). To account for the ability of the SCN's major neurotransmitter (GABA) to be either inhibitory or excitatory (*Albers et al., 2017*), we applied GABAergic synaptic conductances of either polarity ($g_{syn-I}$ and $g_{syn-E}$). Our simulations showed that overall, in the absence of GABAergic synaptic conductance ($g_{Syn-I}$ = 0 nS), $I_A$ led to a suppression of spontaneous firing rate in model SCN neurons (*Figure 7A,B* a1 *vs* a4). This observation is consistent with previous experimental work (*Granados-Fuentes et al., 2012*; *Hermanstyne et al., 2017*). This effect was retained following inclusion of synaptic input of either polarity (*Figure 7A–B*, compare *a2 vs a5, and a3 vs a6*), with the suppressive effect of $g_{syn-I}$ especially augmented by high $I_A$ (*Figure 7B*).

Having observed such effects of $I_A$ on intrinsic activity and cellular response to inputs, we next investigated its effects on the spontaneous activity exhibited by SCN neurons across the circadian day. Here, we simulated the different resting states of *R. pumilio* SCN neurons and day-night changes in spontaneous firing rate (as in the neurons, *Figure 2B,E*, respectively) by subjecting the model to a range of leak currents. Specifically, we varied the scaling factor for the ratio of potassium leak ($g_{LK}$) to sodium leak ($g_{LNa}$) from 0.85 to 1.15 (*Figure 7C*). This was motivated by previous work showing that sodium leak current is higher during the day than at night in mouse SCN neurons (*Flourakis et al., 2015*). Furthermore, it has been suggested that potassium leak currents are lower during the day and higher at night. According to this 'bicycle' model, a $g_{LK}/g_{LNa}$ scaling factor less than 1 corresponds to a daytime 'up-state', and a scaling factor greater than 1 to a night-time 'down-state'. Simulating this variation in leak currents indeed transited the spontaneous RMP and firing rate of the model cells from the daytime depolarized state to night-time suppressed state (as in the neurons, *Figure 2B,G*; *Figure 4—figure supplement 4B–F*). We then tested the influence of $I_A$ on firing rate at each of these electrical states. As reported above, our results revealed that, overall, $I_A$ conductances suppressed spontaneous firing activity (*Figure 7C*, c1-c3), but the extent of this

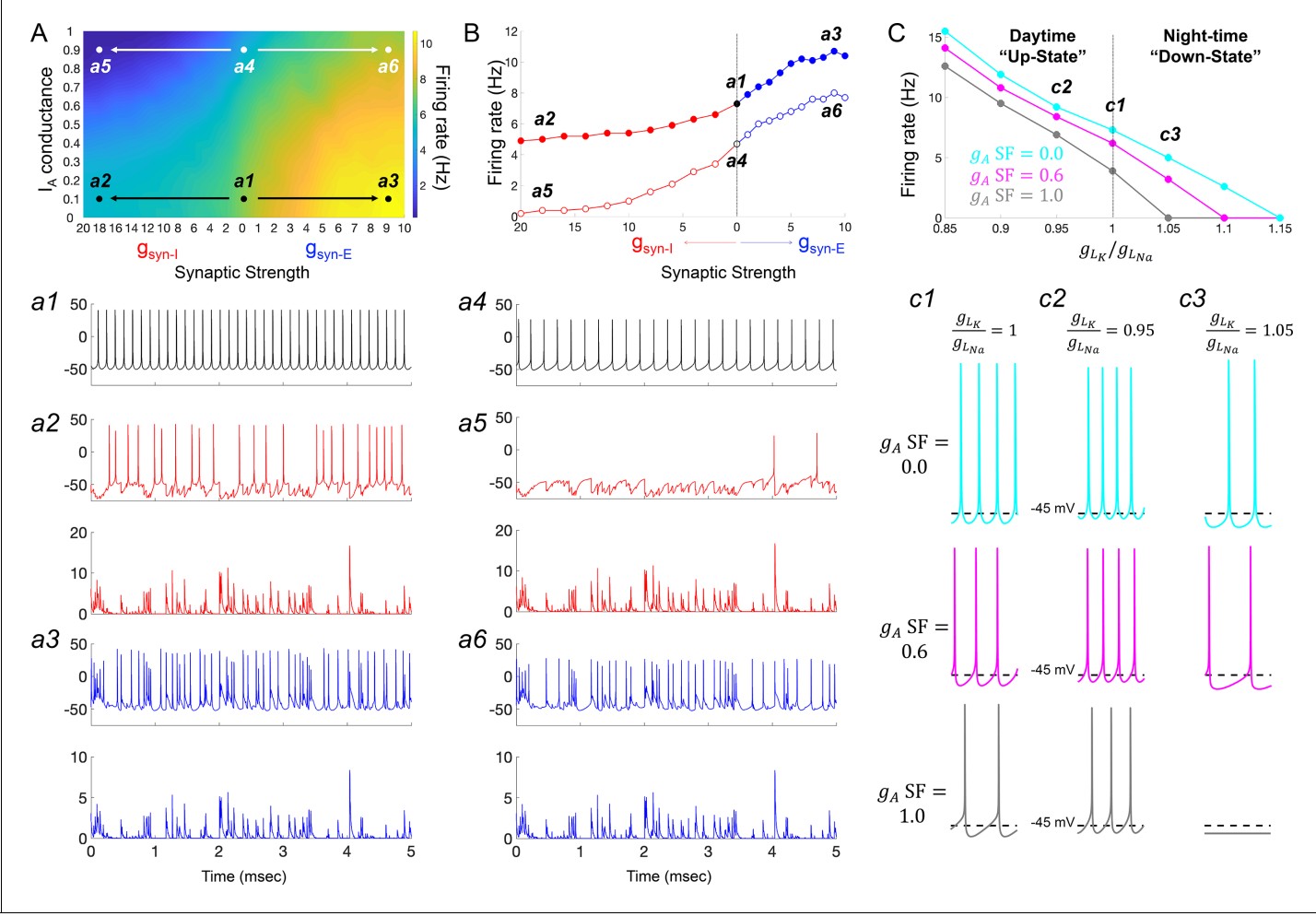

**Figure 7.** $I_A$ conductances act to amplify extrinsic and intrinsic suppressive signals in the *Rhabdomys pumilio* SCN. (A) Heatmap showing the overall effects of inhibitory ($g_{syn-I}$, red) and excitatory ($g_{syn-E}$, blue) physiological GABAergic synaptic conductances on firing frequency with increasing $I_A$ conductances in the model *R. pumilio* SCN. (a1–a3) Examples of firing activity in model cell with low $I_A$ conductance ($g_A$ SF = 0.1) and absence of GABAergic synaptic conductance (a1, $g_{syn-I}$ = $g_{syn-E}$ = 0 nS), high suppressive GABAergic synaptic conductance (a2, $g_{syn-I}$ = 18 nS), or high excitatory GABAergic synaptic conductance (a3, $g_{syn-E}$ = 9). (a4–a6) Examples of firing activity in model *R. pumilio* SCN neurons with high $I_A$ conductance ($g_A$ SF = 0.9) and absence of GABAergic synaptic conductance (a4, $g_{syn-I}$ = 0 nS), high suppressive GABAergic synaptic conductance (a5, $g_{syn-I}$ = 18 nS), or high excitatory GABAergic synaptic conductance (a6, $g_{syn-E}$ = 9). (B) Firing rate as a function of inhibitory ($g_{syn-I}$, red) and excitatory ($g_{syn-E}$, blue) GABAergic synaptic conductances of different strength. Open and filled dots correspond to model cell with high (0.9) or low (0.1) $I_A$ conductance ($g_A$ SF), respectively. (C) Overall effect of intrinsic excitability states (scaling factor for the ratio of potassium leak current ($g_{LK}$) to sodium leak current ($g_{LNa}$) from 0.85 to 1.15) on firing frequency with increasing $I_A$ conductances in the model cell ($g_A$ SF = 0 [cyan], 0.6 [pink] and 1.0 [gray]). $g_{LK}/g_{LNa}$ SF less than one corresponds to a daytime 'up-state', and a SF greater than one to a night-time 'down-state'. (c1) Effect of $I_A$ ($g_A$ = 0, 0.6 and 1.0) on firing rate with nominal potassium/sodium leak current ratio ($g_{LK}/g_{LNa}$ SF = 1). (c2) Effect of $I_A$ ($g_A$ SF = 0, 0.6 and 1.0) on firing rate with reduced potassium/sodium leak current ratio ($g_{LK}/g_{LNa}$ SF = 0.95), representing daytime up-state. (c3) Effect of $I_A$ ($g_A$ SF = 0, 0.6 and 1.0) on firing rate with elevated potassium/ sodium leak current ratio ($g_{LK}/g_{LNa}$ SF = 1.05), representing night-time down-state. Notice that $I_A$ amplifies the suppressive action of the low intrinsic excitability state (during down-state). SF: scaling factor.

The online version of this article includes the following figure supplement(s) for figure 7:

**Figure supplement 1.** Spontaneous synaptic events in *Rhabdomys pumilio* SCN neurons.

suppression was magnified in slow firing and more hyperpolarized cells (*Figure 7C*, c3), such as those frequently recorded at night.

Altogether, these observations are consistent with experimental findings in the SCN, and elsewhere in the brain, that $I_A$ conductances assist suppressive signals. We therefore conclude that in the *R. pumilio* SCN, $I_A$ conductances may act as a 'break' to modulate (tone down) excitation during

the day in depolarized excited cells, and promote inhibition at night in more hyperpolarized slow-firing neurons.

## Discussion

We have applied whole-cell recordings, advanced data assimilation and modeling approaches to provide the first comprehensive description of spontaneous, and evoked, electrical activity of individual SCN neurons in a diurnal species. Our approach reveals strong similarities with the SCN of closely related nocturnal species, but also notable differences.

### Similarities with the nocturnal SCN

Most importantly, the fundamental daily rhythm in electrical excitability ('upstate' during the day and a 'downstate' at night [*Allen et al., 2017*; *Belle and Diekman, 2018*; *Harvey et al., 2020*]) reported for nocturnal species is retained in *R. pumilio*. This reinforces the current view that mechanisms of rhythm generation and regulation are broadly retained across mammalian species with different circadian niches. Moreover, the response of *R. pumilio* SCN neurons to depolarizing inputs and the underlying ionic mechanisms were similar to that of nocturnal rodents (*Belle et al., 2009*). In further support of this view, our modeling revealed similar action potential generation mechanisms in the *R. pumilio* SCN to those in the mouse and rat SCN (*Jackson et al., 2004*; *Belle et al., 2009*).

### Novel properties of the *Rhabdomys pumilio* SCN

The most obvious point of divergence between the *R. pumilio* SCN and that of closely related nocturnal species was its response to hyperpolarizing pulses. Thus, we found that a substantial fraction of *R. pumilio* neurons showed a prominent delay-to-fire (for several hundreds of milliseconds in some cells) following inhibitory pulses. This sort of electrical reaction to inhibition has been observed in neurons elsewhere in the brain (*Schoppa and Westbrook, 1999*; *Saito and Isa, 2000*; *Burdakov and Ashcroft, 2002*; *Burdakov et al., 2004*; *Nadin and Pfaffinger, 2010*; *Tarfa et al., 2017*), but to the best of our knowledge has never before been reported in SCN neurons (*Thomson and West, 1990*; *Pennartz et al., 1998*; *Kuhlman and McMahon, 2004*; *Belle et al., 2009*; *Gamble et al., 2011*; *Belle and Piggins, 2017*). The appearance of such 'Type-B' neurons in the SCN is thus a novel property of *R. pumilio*.

What causes delay-to-fire activity in *R. pumilio* neurons (and why are they absent from the nocturnal SCN)? Our computational models and experimental findings identified the activity of the transient subthreshold A-type potassium channels ($I_A$) as the determinant of this suppressive bioelectrical effect, with the $I_A$ conductance density (which presumably represents the number of functional $I_A$ channels), defining the delay-to-fire latency. The implication, that cells with higher $I_A$ conductances show longer delay-to-fire latencies, finds support from experimental findings elsewhere in the brain (*Schoppa and Westbrook, 1999*; *Saito and Isa, 2000*; *Burdakov and Ashcroft, 2002*; *Burdakov et al., 2004*; *Nadin and Pfaffinger, 2010*; *Tarfa et al., 2017*).

The pore-forming (α) subunits of $I_A$ channels (Kv1.4, 4.1, 4.2 and 4.3) are present in nocturnal rodent (rat, mouse and hamster) SCN neurons, and have been implicated in regulating electrical activity and supporting core clock function (*Huang et al., 1993*; *Bouskila and Dudek, 1995*; *Alvado and Allen, 2008*; *Itri et al., 2010*; *Granados-Fuentes et al., 2012*; *Granados-Fuentes et al., 2015*; *Hermanstyne et al., 2017*). Their failure to produce the delay-to-fire phenotype in those nocturnal species therefore likely reflects some quantitative variation in their function. A likely possibility, consistent both with known features of $I_A$ physiology and our modeling of the *R. pumilio* SCN, is variation in inactivation time constant (timescale over which a channel becomes inactivated following de-inactivation). Elsewhere in the brain it has been shown experimentally that cells expressing $I_A$ channels with faster inactivation time constants (close to 12 ms) show rebound firing, while slower inactivation time constants (~140 ms) produce delay-to-fire activity (*Saito and Isa, 2000*; *Burdakov et al., 2004*). Interestingly, the $I_A$ inactivation time constant measured in mouse and hamster SCN neurons showed relatively fast gating variables (below 22 ms: [*Alvado and Allen, 2008*; *Itri et al., 2010*]), consistent, therefore, with the presence of rebound but not delay-to-fire characteristics in SCN neurons of these species. In agreement, to fully model the range of delay-to-fire behaviors observed in *R. pumilio* SCN neurons, our original mouse model had to be supplemented with $I_A$ channels with a slow inactivation time constant (near 140 ms) (*Figure 6C,O,P*).

Variation in delay-to-fire appeared due to alteration in $I_A$ conductances (*Figure 6M,O*), however, the range of delay latencies observed in our recordings could also be produced by varying the inactivation time constant while holding the $I_A$ conductance constant (*Figure 6N,P*). The inactivation time constants returned by this modeling fall within physiological ranges, and values required to produce delay-to-fire responses are similar to experimentally determined values in other parts of the brain (*Saito and Isa, 2000*; *Burdakov et al., 2004*).

The functional properties of the $I_A$ channel family (Kv4), specifically inactivation time constant and current density, can be influenced by two classes of auxiliary proteins known as Kv channel-interacting proteins (KChIP1–4) and dipeptidyl peptidase-like proteins (DPLPs; DPP6 and DPP10) (*Jerng and Pfaffinger, 2014*). When associated with the various complements of these proteins, the $I_A$ channel inactivation time constant can vary from a few ms to several hundred ms (depending on their expression pattern and the nature of interaction with the channels), reversibly transforming rebound firing to delay firing cells (*Shibata et al., 2000*; *Holmqvist et al., 2002*; *Jerng et al., 2004*; *Jerng et al., 2005*; *Jerng et al., 2007*; *Amarillo et al., 2008*; *Maffie et al., 2009*; *Nadin and Pfaffinger, 2010*). The transcripts for these auxiliary proteins are expressed brain-wide across different mammals, including in the SCN of nocturnal rodents (*Wen et al., 2020*) and the diurnal baboon (*Mure et al., 2018*), and have been implicated in circadian control mechanisms in other excitable cell types (*Jeyaraj et al., 2012*).

A plausible explanation for the range of delay-to-fire activity in the *R. pumilio* SCN, therefore, is variation in activity of KChIP and DPLP proteins producing diversity in inactivation time constants. Interestingly, such a mechanism could also account for the other notably unusual feature of the *R. pumilio* SCN - the absence of a clear relationship between RMP and $R_{input}$ (*Figure 2F*). These $I_A$ auxiliary proteins are known to regulate the input resistance ($R_{input}$) of neurons without changing resting membrane potential (RMP) and capacitance (*Nadin and Pfaffinger, 2010*). Thus, variation in KChIP and DPLP activity across the population of *R. pumilio* SCN neurons could both produce diversity in delay-to-fire activity and disrupt the link between RMP and $R_{input}$ across neurons observed in nocturnal species (*Kuhlman and McMahon, 2004*; *Belle et al., 2009*). Such RMP and $R_{input}$ decoupling may alter the way that the *R. pumilio* SCN integrates synaptic inputs (*Branco et al., 2016*; *Fernandez et al., 2019*).

## Putative functional significance

We applied modeling to determine how $I_A$ channels may regulate excitability in *R. pumilio* SCN neurons in the face of spontaneous (circadian) variations in intrinsic neuronal properties and synaptic input. Experimental results in nocturnal SCN (*Granados-Fuentes et al., 2012*; *Hermanstyne et al., 2017*) and elsewhere in the brain (*Connor and Stevens, 1971*; *Rudy, 1988*; *Liss et al., 2001*; *Baranauskas, 2007*; *Khaliq and Bean, 2008*) reveal that $I_A$ channels can suppress spontaneous firing rate. Our modeling returned a similar impact of $I_A$ in *R. pumilio*, while revealing aspects of this effect that could be especially relevant for a diurnal species. Thus, in general, $I_A$ reduced the effect of intrinsic or synaptically-driven increases in excitability on firing, while enhancing the impact of inhibitory currents (*Figure 7*). The weight of this effect though fell differently across the circadian cycle.

In our model, the weight of the imposed suppression of firing by $I_A$ conductances was stronger at night (in hyperpolarized low-firing neurons) than in the day (in more depolarized fast-firing neurons) (*Figure 7*). In this way, $I_A$ would reinforce the SCN's 'down-state' at night. In nocturnal species, the intrinsic reduction in SCN activity at night is augmented by the appearance of inhibitory inputs associated with activity and arousal at this circadian phase (*van Oosterhout et al., 2012*). Such inhibitory inputs are presumably reduced in diurnal species such as *R. pumilio*, in which activity occurs predominantly during the day. The biophysical properties of $I_A$ channels (conductance active at the subthreshold range of the RMP and progressively becoming available with hyperpolarization), together with its sensitivity to neurotransmitters (*Aghajanian, 1985*; *Yang et al., 2001*; *Burdakov and Ashcroft, 2002*), could provide an opportunity for the *R. pumilio* SCN to compensate for the reduction in inhibitory inputs at night. Accordingly, our modeling evidence favors the interpretation that $I_A$ acts to amplify suppressive signals at night to maintain the low electrical activity in the SCN at this time of day.

The $I_A$ conductance may also be an important response to enhanced excitatory inputs during the day in diurnal species. Day-active animals are exposed to daytime light (the most important excitatory input to the SCN) to an extent that nocturnal species are not (for example, exposure to high

light intensity for longer duration). The ability of $I_A$ to reduce the impact of such excitatory inputs, and perhaps augment the effect of inhibitory inputs from the thalamus, lateral hypothalamus or retina (*Belle et al., 2014*; *Sonoda et al., 2020*) or intrinsic to the SCN (*Hannibal et al., 2010*), would apply an appropriate 'brake' on daytime activity of the SCN.

Our results suggest a potential decoupling between cellular RMP and membrane/input resistance ($R_{input}$) *in R. pumilio* SCN neurons. In the mouse SCN, increased membrane resistance is associated with depolarized RMP during the day, and at night hyperpolarized cells exhibit reduced membrane resistance (*Kuhlman and McMahon, 2004*; *Belle et al., 2009*). In the *R. pumilio* SCN, we found that RMP is more depolarized during the day than at night as in the mouse SCN, but that there is not a significant day-night difference in membrane resistance (*Figure 2F*) due to a subset of day cells with relatively low $R_{input}$ (despite a depolarized RMP) and a subset of night cells with relatively high $R_{input}$ (despite a hyperpolarized RMP). The decoupling of membrane potential and input resistance in the *R. pumilio* SCN could reflect the activity and magnitude of different conductances in this species compared to mouse SCN. In neuronal systems, membrane resistance can determine how cells respond to inputs, with high membrane resistance amplifying synaptic signals (e.g. *Branco et al., 2016*; *Fernandez et al., 2019*). In *R. pumilio*, cells with high membrane resistance during the night therefore may provide an additional cellular mechanism to amplify inhibitory signals. During the day, depolarized cells with low membrane resistance would be less sensitive to excitatory inputs, thereby supporting the daytime 'brake' in extrinsic excitability.

In summary, our whole-cell recordings and computational modeling highlight the potential importance of $I_A$ and cellular membrane resistance in tuning excitability in the *R. pumilio* SCN. This may be an important step in accommodating SCN activity to diurnal living while maintaining the high day/night contrast in electrical activity (RMP and firing rate) necessary to sustain the robustness of the clock, a contributing factor to promote health and wellbeing (*Ramkisoensing and Meijer, 2015*; *Bano-Otalora et al., 2021*).

## Computational modeling approach and considerations

Mammalian neurons, including SCN neurons, possess a wide array of ion channels that contribute to membrane excitability and action potential generation. Our computational model of *R. pumilio* SCN neurons incorporates several of the ionic currents that have been observed in mouse SCN, but some currents are not represented in the model. For example, the large-conductance calcium-activated potassium (BK) channel is known to play a role in circadian variation of SCN excitability (*Whitt et al., 2016*), but we have not included it in our model due to the added complexity involved in modeling intracellular calcium dynamics. Furthermore, the currents that are in the model do not distinguish among the different subtypes of current that exist for each ion. For example, the model contains a single inward calcium current, rather than separate L-, N-, P/Q-, and R-type calcium currents that have distinct activation/inactivation kinetics and are known to be present in SCN neurons (*McNally et al., 2020*). For some currents that are in the model, we also make simplifying assumptions to reduce the number of parameters that need to be estimated. For example, since it is known that the inward sodium current in SCN neurons activates very rapidly, in our model we assume that it activates instantaneously so that we do not have to include parameters associated with the time constant for activation. The lack of a voltage-dependent time constant for sodium activation in our model may explain the subtle difference in the shape of the upstroke of the action potentials in the model compared to the data (*Figure 4C*). In addition, since our model is deterministic, it does not capture the irregularity in spike timing or the small voltage fluctuations that are present in the recordings due to ion channel noise or synaptic input (e.g. see *Figure 4—figure supplement 2*).

## Applying the data assimilation method to physiology

Our results demonstrate that data assimilation (DA) is a powerful tool for developing conductance-based models. Our state-of-the-art DA algorithm was able to reliably perform state and parameter estimation for *R. pumilio* SCN neuron models from current-clamp recordings without the use of voltage-clamp and pharmacological agents to isolate specific currents, and without the injection of custom-designed stimulus waveforms as used in other DA approaches (*Meliza et al., 2014*). Rather, we made judicious use of the voltage traces resulting from standard depolarizing and hyperpolarizing current steps. This is an important step forward for the practicality of applying DA methodology in

the neuroscience context, as it enables model-building from the plethora of past, present, and future current-clamp recordings obtained by electrophysiology labs using classical current-step protocols.

## Materials and methods

### Animals

All animal use was in accordance with the UK Animals, Scientific Procedures Act of 1986, and was approved by the University of Manchester Ethics committee. Adult *R. pumilio* (male and female, age 3–9 months) were housed under a 12:12 hr light dark cycle (14.80 Log Effective photon flux/cm$^2$/s for melanopsin or Melanopic EDI (equivalent daylight illuminance) of 1941.7 lx) and 22°C ambient temperature in light tight cabinets. Lighting conditions were aimed to reproduce the *R. pumilio* experience of natural daylight by approximating the relative activation for melanopsin, rod opsin, and cone opsins (*Bano-Otalora et al., 2021*). Food and water were available ad libitum. Cages were equipped with running wheels for environmental enrichment. Zeitgeber Time (ZT) 0 corresponds to the time of lights on, and ZT12 to lights off. Original *R. pumilio* breeding pairs used to establish our colony were kindly provided by the Hoekstra lab at Harvard University.

### Brain slice preparation for electrophysiological recordings

Following sedation with isoflurane (Abbott Laboratories), animals were culled by cervical dislocation during the light phase (beginning of the day (n=4) or late day (n=4)). Brains were immediately removed and mounted onto a metal stage. Brain slices were prepared as described previously (*Hanna et al., 2017*). 250 µm coronal slices containing mid-SCN levels across the rostro-caudal axis were cut using a Campden 7000smz-2 vibrating microtome (Campden Instruments, Loughborough, UK). Slices were cut in an ice-cold (4°C) sucrose-based incubation solution containing the following (in mM): 3 KCl, 1.25 NaH$_2$PO$_4$, 0.1 CaCl$_2$, 5 MgSO$_4$, 26 NaHCO$_3$, 10 D-glucose, 189 sucrose, oxygenated with 95% O$_2$, 5%CO$_2$. After slicing, tissue was left to recover at room temperature in a holding chamber with continuously gassed incubation solution for at least 20 min before transferring into recording aCSF. Recording aCSF has the following composition (mM): 124 NaCl, 3 KCl, 24 NaHCO$_3$, 1.25 NaH$_2$PO$_4$, 1 MgSO$_4$, 10 D-Glucose and 2 CaCl$_2$, and 0 sucrose; measured osmolarity of 300–310 mOsmol/kg. Slices were allowed to rest for at least 90 min before starting electrophysiological recordings.

### Whole-cell patch clamp recordings

SCN brain slice electrophysiology was performed as previously described (*Belle et al., 2014*). SCN coronal brain slices were placed in the bath chamber of an upright Leica epi-fluorescence microscope (DMLFS; Leica Microsystems Ltd) equipped with infra-red video-enhanced differential interference contrast (IR/DIC) optics. Brain slices were kept in place with an anchor grid, and continuously perfused with aCSF by gravity (~2.5 ml/min). Stock solution for 4-Aminopyridin (4-AP; Cat. No. 0940, Tocris, Bristol, UK) was made in aCSF. Working 4-AP dilution (5 mM) was made in aCSF immediately before bath application. Recordings were performed from neurons located across the whole SCN during the day (ZT4 to ZT12) and at night (ZT13 to ZT22; *Figure 2A*). SCN neurons were identified and targeted using a 40x water immersion UV objective (HCX APO; Leica) and a cooled Teledyne Photometrics camera (Retiga Electro), specifically designed for whole-cell electrophysiology. Photographs of the patch pipette sealed to SCN neurons were taken at the end of each recording for accurate confirmation of anatomical location of the recorded cell within the SCN.

Patch pipettes (resistance 7–10 MΩ) were fashioned from thick-walled borosilicate glass capillaries (Harvard Apparatus) pulled using a two-stage micropipette puller (PB-10; Narishige). Recording pipettes were filled with an intracellular solution containing the following (in mM): 120 K-gluconate, 20 KCl, 2 MgCl$_2$, 2 K$_2$-ATP, 0.5 Na-GTP, 10 HEPES, and 0.5 EGTA, pH adjusted to 7.3 with KOH, measured osmolarity 295–300 mOsmol/kg.

An Axopatch Multiclamp 700A amplifier (Molecular Devices) was used for voltage-clamp and current-clamp recordings. The electrode offset potential was compensated before establishing a seal and the liquid junction potential was not corrected. Signals were sampled at 25 kHz and appropriately acquired in gap-free or episodic stimulation mode using pClamp 10.7 (Molecular Devices). Series resistance (typically 10–30 MΩ) was corrected using bridge-balance in current-clamp

experiments and was not compensated during voltage-clamp recordings. Access resistance for the cells used for analysis was <30 MΩ. Post-synaptic currents (PSCs) were measured under voltage-clamp mode while holding the cells at −70 mV. Measurement of spontaneous activity in current-clamp mode was performed with no holding current (I=0). All data acquisition and protocols were generated through a Digidata 1322A interface (Molecular Devices). Recordings were performed at room temperature (~ 23°C). A portion of the data appearing in this study also contributed to the investigation of the impact of daytime light intensity on the neurophysiological activity and circadian amplitude in the *R. pumilio* SCN (*Bano-Otalora et al., 2021*).

## Membrane properties of SCN neurons

Resting membrane potential (RMP), spontaneous firing rate (SFR) and input resistance ($R_{input}$) were determined within 5 min of membrane rupture. Average SFR in firing cells was calculated as the number of action potentials per second within a 30 s window of stable firing using a custom-written Spike2 script, and average RMP was measured as the mean voltage over a 30 s window. $R_{input}$ was estimated using Ohm's law (R=V/I) where V represents the change in voltage induced by a hyperpolarizing current pulse (−30 pA for 500 ms) as previously described (*Belle et al., 2009*). The neurone's response to excitatory and inhibitory stimuli was identified by a series of depolarizing and hyperpolarizing current pulses (from −30 to +30 pA in 5 pA steps, duration 1 s).

## Immunohistochemistry

*R. pumilio* were culled during the light phase and brains were fixed in 4% PFA, followed by 5 days in 30% sucrose. 35 µm brain sections were cut using a freezing sledge microtome (Bright Instruments, Huntingdon, UK). Immunofluorescence staining was performed as previously described (*Timothy et al., 2018*). Briefly, slices were washed in 0.1M PBS and 0.1% TritonX-100 in PBS before incubation with blocking solution (5% donkey serum (Jackson ImmunoResearch, Pennsylvania, US) in 0.05% Triton-X100 in 0.1M PBS). After 60 min, sections were incubated for 48 hr at 4°C with primary antibodies (AVP Rabbit, Millipore AB1565, 1:5000; VIP Rabbit, Enzo, VA1280-0100, 1:1000; GRP Rabbit, Enzo GA1166-0100, 1:5000). Following washes, slices were incubated overnight with secondary antibodies (1:800; Donkey anti-rabbit Cy3, Jackson ImmunoResearch). Slices were finally mounted onto gelatine coated slides and cover-slipped using DAPI-containing Vectashield anti-fade media (Vector Laboratories, Peterborough, UK). Digital photos were taking using a Leica DFC365 FX camera connected to a Leica DM2500 microscope using Leica Microsystems LAS AF6000 software.

## Data analysis

Current-clamp data were analyzed using Spike2 software (Cambridge Electronic Design, CED). Data analysis was performed by experimenters blinded to the time-of-day when neurons were recorded. Since multiple neurons were recorded from a single slice (a total of 111 neurons from eight animals), electrophysiological data (RMP, SFR, and $R_{input}$) were compared using a multilevel mixed-effects linear model that included the slice that each cell was recorded from as a random effect and the time-of-day as fixed effect. 4-AP effect on delay to fire behavior was analyzed using one-way repeated measures ANOVA followed by Tukey's *post hoc* test. All statistical analyses were performed using SPSS version 23 (SPSS Inc, Chicago, IL, USA) and GraphPad Prism 7.04 (GraphPad Software Inc, CA, USA). For all tests, statistical significance was set at $p < 0.05$. Data are expressed as mean ± SEM. Sample sizes are indicated throughout the text and figure legends. The number of replications (n, number of data points used in the statistical tests) is the number of neurons recorded. Sample sizes were based on our previous publications and work of others (*Pennartz et al., 1998*; *Belle et al., 2009*; *Hermanstyne et al., 2016*; *Timothy et al., 2018*). Percentages of cells in the different electrophysiological states and responses to depolarizing and hyperpolarizing pulses during the day and at night were analyzed using Chi-Squared test.

## Model estimation strategy

Traditionally, conductance-based (or Hodgkin-Huxley-type) models of neurons are constructed using voltage-clamp (VC) measurements of individual ionic currents. While VC can provide accurate descriptions of certain channel properties, its execution is experimentally labour intensive, and by measuring each current in isolation VC protocols do not capture the dynamical interplay between

the many active channels that drive complex and integrated electrical behaviors in mammalian neurons. Furthermore, it is not feasible to use VC to measure all the ionic currents of interest from the same cell, due to the limited amount of time available to perform patch-clamp recordings before the cell dialyzes (approximately 5–10 min) and the need to wash out the pharmacological agents used to isolate and measure one current before isolating and measuring the next. Thus, a model constructed using VC data is not a representation of the currents active in a single cell, but rather is a combination of currents measured across several different cells (*Golowasch et al., 2002*).

The advantage of current-clamp (CC) protocols is that the recorded voltage trace reflects the natural interaction of all the ionic conductances within that cell. The challenge for constructing a model based on CC data is that only one of the state variables of the model, the membrane voltage, has been measured directly; the gating variables that represent the opening and closing of ion channels are unobserved. Each ionic current has several parameters associated with it that are typically not known a priori and must also be estimated from the data.

Data assimilation is widely used in fields such as geoscience and numerical weather prediction but has only recently begun to be applied in neuroscience. One of the main classes of DA algorithms are variational methods such as 4D-Var that identify optimal solutions over a time window of measurements and are able to deal effectively with a large number of unobserved state variables and unknown parameters. Since our *R. pumilio* SCN model has many parameters that are not known a priori we chose to employ the variational approach in this study.

The variational algorithm strongly constrains the estimated dynamical state of the neuron to conform with model expectations, while penalizing deviations from measurements using a least-squared error metric. The constrained optimization problem is regularized using a 'nudging' term to push estimates of the voltage towards the data. We used current-clamp data from multiple protocols (*Figure 4—figure supplement 1*) simultaneously to inform the estimated model of the cell's characteristic responses to changes in the applied current. We initially used a set of channels in our *R. pumilio* model similar to that previously used for a mouse SCN model (*Belle et al., 2009*; *Figure 4A*), but permitted each of the parameters in the model the freedom to be distinct for each individual cell that we fit. We started the estimation algorithm for each cell using over 50 initial guesses for the parameters and state variables. We performed model selection by assessing a Pareto frontier consisting of the DA cost function evaluation and the mismatch in firing rate between the data and simulations of the resulting model under various current-clamp conditions. These simulations were performed using the ode15s and ode45 solvers in MATLAB.

## Conductance-based model

In the original version of the mouse SCN model (*Sim and Forger, 2007*; *Belle et al., 2009*), the parameter values are spread over a wide range and the gating variable expressions are not uniform, which creates complications when constructing our optimization problem. We aimed to fit to current-clamp data of the *R. pumilio* using the same set of currents as the mouse model but expressing their kinetics uniformly. Additionally, we separated the leak into sodium and potassium components to investigate the role each may play in altering the resting membrane potential of cells in day versus night, as was done in *Diekman et al., 2013*. Lastly, we approximated the sodium and A-type activations as instantaneous, as has been done previously to reduce the dimensionality of the SCN model (*Sim and Forger, 2007*). Conversely, we allowed the inactivation of sodium to have a wide range of permissible time constant values, as persistent sodium is known to play a role in maintaining the pace of firing (*Harvey et al., 2020*). Thus, our sodium channel functionally played the classical role of a transient sodium current in generating the upstroke of the action-potential, but also is possibly involved in governing certain subthreshold properties. The full model is described by the following equations:

$$C\frac{dV}{dt} = I_{app}(t) - I_{Na} - I_K - I_{Ca} - I_{L_{Na}} - I_{L_K} - I_H - I_A - I_{syne-E} - I_{syn-I}$$

$$= I_{app}(t) - g_{Na}m_{Na_\infty}^3 h_{Na}(V - E_{Na}) - g_K n^4(V - E_K) - g_{Ca}m_{Ca}h_{Ca}(V - E_{Ca}) - g_{L_{Na}}(V - E_{Na})$$
$$- g_{L_K}(V - E_K) - g_H m_H(V - E_H) - g_A m_{A_\infty}^3 h_A(V - E_K)$$

$$-g_{syne-E}s(t)\left(V-E_{syn-E}\right)-g_{syn-I}s(t)\left(V-E_{syn-I}\right)$$

$$\frac{dq}{dt}=\frac{q_{\infty}(V)-q}{\tau_q(V)}, q=\{m_i,h_i,n\}$$

$$q_{\infty}(V)=\frac{1}{2}+\frac{1}{2}\tanh\left(\frac{V-v_q}{dv_q}\right)$$

$$\tau_q(V)=\tau_{q0}+\tau_{q1}\left(1-\tanh^2\left(\frac{V-v_q}{dv_q}\right)\right)$$

where $C$ is membrane capacitance, $V$ is membrane potential, $I_{app}(t)$ is the applied current, $I$ are ionic currents, $g$ are maximal conductances, $E$ are reversal potentials, and $q$ are gating variables with steady-state functions $q_{\infty}$ and time constants $\tau_q$. The active conductance of a channel, $G$, is the product of its maximal conductance and gating variables, e.g. $G_{Na}=m_{Na_{\infty}}^3 h_{Na}$. The $g_A$ and $\tau_{h_A}$ scaling factors used in *Figures 6* and *7* are coefficients that multiply the maximal conductance parameter and time constant variable, respectively. The scaling factor for the ratio of potassium to sodium leak conductance used in *Figure 7* and *Figure 4—figure supplement 4* is a coefficient that divides $g_{L_{Na}}$ and multiplies $g_{L_K}$. We calculated the synaptic gating variable $s(t)$ from voltage-clamp recordings of postsynaptic currents in *R. pumilio* SCN neurons with the cells held at -70 mV. The synaptic currents $I_{syn-E}$ and $I_{syn-I}$ were not used in the DA procedure, and were only included in the model simulations shown in *Figure 7A-B*. The $I_H$ and $I_A$ currents were only included in the DA procedure and model simulations shown in *Figures 6B-C* and *7*. The parameter values used for all model simulations are provided in the *Supplementary file 1*.

# Acknowledgements

We thank the members of the University of Manchester Biological Services Facility for their excellent assistance in colony maintenance and husbandry. We also thank Profs Luckman and Randall for allowing us access to their electrophysiology equipment. This work was funded by a Biotechnology and Biological Sciences Research Council (BBSRC) Industrial Partnership Award with Signify (BB/P009182/1) to RJL, and by grants from the BBSRC to MDCB (BB/S01764X/1) and to TMB (BB/N014901/1), and the Wellcome Trust (210684/Z/18/Z) to RJL. This material is based upon work supported by grants from the National Science Foundation (DMS 155237), the US Army Research Office (W911NF-16-1-0584), and the US-UK Fulbright Commission to COD. MJM and COD gratefully acknowledge the financial support of the EPSRC via grant EP/N014391/1.

# Additional information

### Competing interests

Timothy Brown, Robert J Lucas: has received investigator-initiated grant funding from Signify (formerly Philips Lighting). The other authors declare that no competing interests exist.

### Funding

| Funder | Grant reference number | Author |
| --- | --- | --- |
| Biotechnology and Biological Sciences Research Council | BB/P009182/1 | Timothy Brown Robert J Lucas |
| Biotechnology and Biological Sciences Research Council | BB/S01764X/1 | Mino DC Belle |
| Biotechnology and Biological Sciences Research Council | BB/N014901/1 | Timothy Brown |
| Wellcome Trust | 210684/Z/18/Z | Robert J Lucas |

| National Science Foundation | DMS 155237 | Casey O Diekman |
| --- | --- | --- |
| Army Research Office | W911NF-16-1-0584 | Casey O Diekman |
| US-UK Fulbright Commission | | Casey O Diekman |
| Engineering and Physical Sciences Research Council | EP/N014391/1 | Matthew J Moye Casey O Diekman |

The funders had no role in study design, data collection and interpretation, or the decision to submit the work for publication.

### Author contributions

Beatriz Bano-Otalora, Conceptualization, Formal analysis, Writing - original draft, Writing - review and editing, Performed electrophysiology experiments; Matthew J Moye, Conceptualization, Formal analysis, Writing - original draft, Writing - review and editing, Performed mathematical modelling; Timothy Brown, Robert J Lucas, Conceptualization, Funding acquisition, Writing - review and editing; Casey O Diekman, Conceptualization, Formal analysis, Funding acquisition, Writing - original draft, Writing - review and editing, Performed mathematical modelling; Mino DC Belle, Conceptualization, Formal analysis, Funding acquisition, Writing - original draft, Writing - review and editing, Performed electrophysiology experiments

### Author ORCIDs

Beatriz Bano-Otalora http://orcid.org/0000-0003-4694-9943
Robert J Lucas https://orcid.org/0000-0002-1088-8029
Casey O Diekman https://orcid.org/0000-0002-4711-1395
Mino DC Belle https://orcid.org/0000-0002-4917-957X

### Ethics

Animal experimentation: All animal use was in accordance with the UK Animals, Scientific Procedures Act of 1986, (project licence number PPL 70/8918) and was approved by the University of Manchester Ethics committee.

### Decision letter and Author response

Decision letter https://doi.org/10.7554/eLife.68179.sa1
Author response https://doi.org/10.7554/eLife.68179.sa2

## Additional files

### Supplementary files

• Supplementary file 1. Parameter values for the computational models of *Rhabdomys pumilio* SCN neurons. Models were fit to data from seven different neurons, including a non-adapting cell (*Figure 5Ai*), an adapting-firing cell (*Figure 5Aii*), an adapting-silent cell (*Figure 5Aiii*), two Type-A rebound spiking cells (*Figures 4* and *6A*), and two Type-B delay cells (*Figure 6B and O*). These cells were chosen for modeling because they are representative of the various responses observed across all the recordings. Each model was fit to data from six voltages traces as illustrated in *Figure 4—figure supplement 1*.

• Transparent reporting form

### Data availability

All data generated or analysed during this study are included in the manuscript and supporting files. Source data files have been provided for Figures 2, 3 and 6. Code for simulating our conductance-based models is available in ModelDB (McDougal et al 2017, J Comput Neurosci) at http://modeldb.yale.edu/267183. Code for performing neuronal data assimilation (neuroDA) to infer model parameters from current-clamp recordings is available at https://github.com/mattmoye/neuroDA;

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

## Appendix 1

### Data assimilation algorithm

Here, we briefly describe the variational DA algorithm employed in this paper (see *Moye, 2020* for further details). We represent the neuronal recordings using the following state-space description:

$$x_{k+1} = f(x_k) + \omega_{k+1}, \ \ x_k \in R^L$$

$$y_{k+1} = V_{k+1} + \eta_{k+1}, \ \ y_k \in R^1$$

where $x_k$ is interpreted as the state of the neuron at some time $t_k$, and $y_k$ are our observations (i.e. the voltage measurements). The random variables $\omega_k$ and $\eta_k$ represent model error and measurement error, respectively. We assume that $\omega_k \sim \mathfrak{N}(0, Q)$ and $\eta_k \sim \mathfrak{N}(0, R)$, where $Q$ and $R$ are the model error and measurement error covariance matrices, and that these have no cross-covariance.

*Strong 4d-Var* forces our observations to be consistent with the model $f$. This can be considered the result of taking $Q \to 0$, which yields the nonlinearly constrained problem:

$$C(x) = \frac{1}{2} \sum_{k=0}^{N} R^{-1}(y_k - V_k)^2$$

such that

$$x_{k+1} = f(x_k), \ \ k = 0 \dots N$$

where $R^{-1}$ can now be scaled out completely.

In the cost function, the estimated voltage is expected to be consistent with the dynamics for large model weighting $Q^{-1}$, but the dynamics cannot possibly reproduce the irregularity in the data.

Dynamical State and Parameter Estimation (DSPE) is a technique described by *Abarbanel et al., 2009*, with the premise being to stabilize the synchronization manifold of data assimilation problems by adding a control or "nudging" term $u$. The cost function then becomes:

$$C(x) = \frac{1}{2} \sum_{k=0}^{N} R^{-1}(y_k - V_k)^2 + \sum_{k=0}^{N} u_k^2$$

This synchronization procedure has also been considered for specific functional forms of $u$ in the neuroscience context in *Brookings et al., 2014* wherein they set up an optimal search strategy applied to real data. The nudging strategy in general has been used in geosciences primarily for state estimation (*Park and Xu, 2013*). As shown in *Toth et al., 2011* and *Abarbanel et al., 2009*, the control $u$ acts to reduce conditional Lyapunov exponents.

The goal of DSPE is to define a high-dimensional cost functional which weakly constrains the estimated states to the system observations, and strongly constrains the estimates to the controlled model dynamics while penalizing the control. Without the control, the problem is explicitly formulated as a strong constraint 4D-Var. However, the basin of attraction for global minima along the optimization manifold is shallow. Also, while the minimization term itself is convex, the nonlinearities present in the model constraints generate a large degree of non-convexity in the solution manifold. The intended effect of the nudging term is to smooth the surface. Given that the system is so high dimensional and tightly coupled, formally visualizing this surface is not achievable for our parameter estimation problems.

In the DSPE framework, parameters and states at each point in time are taken on equal footing. Namely, the solution space of the cost function is $(L+1)(N+1) + D$ where $D$ is the number of fixed parameters to infer and $L$ is the number of dynamical variables. Additionally, we are solving for the control $u(t)$ at each point in time. The control is penalized quadratically in an effort to reduce the impact of it at the end of the optimization procedure. While having the control present enforces the data in the model equations, by minimizing it, one is attempting to recover back the minima subject to the uncontrolled model of the system. So, as $u \to 0$ over the course of the optimization, the physical system strong constraint is recovered. We note that in the results presented here, the control term was not fully eliminated by the end of the assimilation window. This may be due to intrinsic

voltage-gated conductances present in the cell that are not included in our model, or other factors such as synaptic input or channel noise.

We must choose a particular transcription method to prescribe our equality constraints. We define our state vector as $x = (V, \bar{x})$ and our uncontrolled dynamics as:

$$\frac{dx}{dt} = f_x(x, \theta)$$

where we can separate out the terms with observations. We assume we only have observations of the voltage of one cell in one compartment (with natural generalizations to networks and multi-compartment descriptions):

$$\frac{dV}{dt} = f_V(\bar{x}, V; \theta)$$

$$\frac{d\bar{x}}{dt} = f_{\bar{x}}(\bar{x}, V; \theta).$$

Then our controlled dynamics become

$$\frac{dV}{dt} = f_V(\bar{x}, V; \theta) + \mathrm{u}(V_{Obs} - V)$$

$$\frac{d\bar{x}}{dt} = f_{\bar{x}}(\bar{x}, V; \theta)$$

where it is understood that $u(t)$ appears only at observational times.

We can formulate the constraints using either a multiple-shooting style approach or using collocation. We will assume measurements are taken uniformly at $t_k = t_0 + k\tau_{obs}$. High resolution measurements are preferred so that we can have control and knowledge of the system at basically every knot point. However, there are circumstances where we may not have data with that level of precision, or we may desire to downsample our data. For that reason, we will say that we have a set of times upon which our constraint equations are satisfied, namely $t_m = t_0 + m\tau_{col}$ where we simply require that the ratio of these time differences is a positive integer, $\frac{\tau_{obs}}{\tau_{col}} \in N$.

To reiterate, the constraints are what connect each of our time points $[t_m, t_{m+1}]$ to one another.

We use a direct collocation method due to the stability options afforded to us for our highly complex, nonlinear problem. With collocation, implementation of implicit methods is effectively as simple as explicit methods. We choose to use Hermite-Simpson collocation which approximates the set of discrete integrations using Simpson's rule. We introduce midpoints in this fashion $\left(x_{k+\frac{1}{2}}\right)$, which are approximated using Hermite interpolation.

$$x_{k+1} - x_k = \frac{1}{6}h_k\left(f_k + 4f_{k+\frac{1}{2}} + f_{k+1}\right)$$

$$x_{k+\frac{1}{2}} = \frac{1}{2}(x_k + x_{k+1}) + \frac{h_k}{8}(f_k - f_{k+1})$$

where $f_k = f_x(x_k, \theta)$ and $\theta$ is the present estimate of $\theta$ constant across our time window.

Here, we take the midpoint and endpoint conditions on equivalent footing for our constraints, $g(x) = 0$, in what is known as its 'separated form'. Therefore, we implement these equations so that $h_k = 2\tau_{col}$ based upon our previous notation, and we have $LN$ equality constraints.

## State and parameter bounds

Setting lower and upper bounds for the state and parameter estimates, $x^L \leq x \leq x^U$, can improve the performance of the DA algorithm. For the states, we specify that the voltage is within a plausible physiological range based on prior knowledge of the system and the variance in the observations. The gating variables are restricted to their dynamic range between 0 and 1. As for the parameters, it

is difficult to know how tight the boundaries should be. As a rule of thumb, if it is possible to parameterize the model in a systematic and symmetric way, it may be easier to construct meaningful bounds. Also, it is advisable to keep the parameters within a bounding box which prevents blow-up of the dynamics such as divisions by zero. The maximal conductances are positive valued, and the sign of the slope for the steady-state gating functions should dictate if they are activating (positive) or inactivating (negative).

Parameter bounds for passive properties of the system, such as cell capacitance and ionic reversal potentials, can be set based on background electrophysiological knowledge or voltage-clamp data and other measurements.

## Implementation

We have implemented 4D-Var in MATLAB using the CasADi framework, (*Andersson et al., 2019*). The 'cas' comes from 'computer algebra system', in which the implementation of mathematical expressions resembles that of any other symbolic toolbox, and the 'AD' for algorithmic (automatic) differentiation. These expressions are then easily used for generating derivatives by breaking the expressions into a number of atomic operations with explicit chain rules, with natural extensions to vector and matrix-valued functions. CasADi data types are all sparse matrices, and low-level scalar expressions (SX type) are stored as directed acyclic graphs where their numerical evaluation is conducted using virtual machines. For nonlinear programming problems, matrix expressions (MX type) are constructed to form the structure of the nonlinear program e.g. the collocation expression. The low-level expressions, e.g. the differential equations, are built using SX type to create a hierarchy of functions for evaluation efficiency and memory management. CasADi will generate the gradient and Hessian information through AD which are then passed to the solver of choice. We elect to solve the optimization problem with IPOPT (Interior Point OPTimize) (*Wächter and Biegler, 2006*). The high-dimensional linear algebra calculations are done using the linear solver MUMPS (MUltifrontal Massively Parallel sparse direct Solver) which is readily distributed with CasADi and interfaced with IPOPT.

## Downsampling

We utilized a downsampling strategy on the current-clamp data in order to facilitate the use of longer stretches of data without exceeding the computational limits on the size of the optimization problem that our computing resources can handle. We set a threshold of $-20$ mV for each action potential, and within a region of 30 ms on either side of when this threshold is hit, the full 25 kHz sampling is preserved. Outside of this window, the data used is downsampled by a factor. For the results presented here, we used a downsampling factor of 5 so that during the action potential the resolution is 25 kHz and outside the time window of the action potential it is 5 kHz. With this strategy, we can retain as many data points as possible during the action potential, which occurs on a much faster timescale than the membrane dynamics during the interspike interval, and this enables us to better fit the spike shape. We also used the full 25 kHz sampling for the 30 ms region immediately following the onset or offset of the depolarizing and hyperpolarizing pulses.

## Multiple observations

A novel component of our DA approach is the use of multiple observations to inform a unified model for each cell's electrophysiology. While the underlying matrices associated with the nonlinear optimization problem are highly sparse, utilizing a desirably large amount of data in combination with the complexity of the equations results in a bottleneck with regards to computer RAM. Therefore, we are restricted through a computational and physical memory budget with regard to our implementation on the amount of data we can use for each estimation. In a sense, we simultaneously solve a series of variational sub-problems that are connected through mutually shared parameters. We use symmetric current-clamp protocols which start with spontaneous activity, followed by either a depolarizing or hyperpolarizing step for 1 s, and a subsequent return to spontaneous activity. A period of a few hundred ms prior to two different hyperpolarizing steps is used so as to access leak channel information and transient inactivation profiles. We use two similar segments from the return from hyperpolarizing steps to inform de-inactivation and activation time scales from rest. Similar

data for two responses to depolarizing steps is used to characterize the firing profiles and understand the limiting behaviour for high-amplitude depolarizing pulses, including regular firing, firing with adaptation, or silence. We bias the data with a large segment (1500 ms) of data during spontaneous activity to reproduce the hallmark spontaneous activity and spike shape of SCN neurons in our estimated models. In the problem construction shown by *Figure 4—figure supplement 1*, 4.5 s of data in total are used for the assimilation, amounting to around 36,000 time points after incorporating our downsampling strategy.

