## [Decision Letter]

**Acceptance summary:**

The manuscript has two novel elements that will be particularly interesting to computational neurobiologists, electrophysiologists and chronobiologists. The first part of the work uses advanced data assimilation and modeling approaches to develop computational models of SCN neuronal activity from current-clamp recordings. The second novel element is the use of a diurnal mouse model and whether there are differences in the functioning of SCN neurons in diurnal versus nocturnal animals. The results that the SCNs function very similarly is not unexpected but adds to a body of work that the timing of the molecular clock and the electrical activity of SCN neurons are similar in diurnal and nocturnal animals. The "switch" for diurnal and nocturnal timing is probably located downstream in the hypothalamus.

**Decision letter after peer review:**

Thank you for submitting your article "Daily electrical activity in the master circadian clock of a diurnal mammal" for consideration by *eLife*. Your article has been reviewed by 3 peer reviewers, and the evaluation has been overseen by a Reviewing Editor and Catherine Dulac as the Senior Editor. The following individual involved in review of your submission has agreed to reveal their identity: Charles N. Allen (Reviewer #3).

While we think that your study is already quite strong, several important issues have been raised by each of the reviewers. We think that all of them are reasonable and valuable and hence critical to appropriately address. Thus, you don't find a separate summary list here, but please check and respond to each point in the three "Recommendation for the authors" sections.

*Reviewer #1 (Recommendations for the authors):*

The major aim of the work was to shed light on potential differences in electrophysiological dynamics in the SCN of diurnal and nocturnal mammals. Bano-Otalora et al., provide insights on the electrophysiological dynamics of the mammalian brain, which knowledge often derives from the investigation of nocturnal animals. In this work, the authors focus on the diurnal rodent Rabdomys pumilio, taking advantage of whole-cell recording (first time in a diurnal mammal), and cutting-edge data assimilation/mathematical modelling to shed light on unexplored properties of a diurnal brain. More in detail, with this approach the authors identified (1) the non-correlation between R input and resting membrane potential (RMP), and (2) rebound hyperpolarization and the consequent delay-to-fire, as substantial differences between R. pumilio and mouse SCN. With these discovers, the authors clearly achieved their aims, and the conclusions are well supported by the combination of data presented and literature cited. However, in spite of a robust experimental design, little to no space has been dedicated to connect the relatively "minor" differences in the properties of R. pumilio SCN compared to mouse and the markedly different lifestyles of diurnal and nocturnal rodents. In other words, how the few peculiarities identified contributes to the accommodation of a dramatically different temporal niche remains unclear. Moreover, the predictions deriving from the mathematical modelling are not proved experimentally at any level, and this compromise the strength of the conclusions, which in some points result more as pure speculations.

In spite of this, the first whole-cell recording of SCN neurons from a diurnal species, represents definitely a necessary step to better understand the neurophysiological scenario potentially allowing the accommodation of a diurnal niche. Although their contribution to the niche switch remains elusive, the peculiarities observed represent anyway the firsts (at least this is what emerges from the authors) differences in the electrophysiological properties of diurnal brains, and could not be identified using the methodology employed in previous papers focusing on diurnal rodents. Finally, the data assimilation and mathematical modelling developed represents an intriguing approach to complement in vivo strategies (especially the investigation of specific electrophysiological features technically challenging), but should be used to generate models that can be somehow (and at least partially) validated in vivo.

Overall, I consider this work worthy of publication in *eLife*, yet I ask the authors to clarify/address the following points:

– Why do rodents have diurnal peaks of spontaneous electrical activity (and in general SCN more exited) irrespectively from their diurnal or nocturnal niche? I know this is not the main aim of the work, but I think that the clarity of the entire manuscript will certainly benefit from specifying this. Especially for non experts, this omission generates expectations that are not fulfilled by the findings provided, partially devaluating the work itself.

– I personally found the description of the aims and the impact too hasty and vague. For example, knowing already that the overall dynamics are preserved between nocturnal and diurnal species (SCN neurons are overall more exited during the day), why did the authors hypothesize a detailed characterization of the whole-cell electrophysiological dynamics could unravel the mysteries of the accommodation of mammalian diurnal niches? I suggest to be more specific since the very beginning on why this approach can help to understand how a diurnal niche could have been accomodated.

– In line with the last comment, I would like the authors to be more open on the current hypothesis suggested to explain diurnal/nocturnal niche accommodation. Given the literature, probably differences in the electrophysiological dynamics is not the only one. Then they could justify why this is the most likely to them, and thus it deserves to be investigated in depth. To do this, I would start asking: how can species adapt to diurnal or nocturnal niches?

– In the main text, why VIP, AVP and GRP peptides have been selected is not clear. Only in Figure 1 legend the authors refer to them as "main SCN neuropeptides". I would suggest the authors to better communicate the choice of these neuropeptides in the main text, perhaps providing also one sentence about their function (i.e. circadian regulation, sleep/wake, etc…).

– Considering Figure 2, it is not clear to me why the authors first described the spontaneous excitability states in Figure 2B (5 states), and then they report only 3 states in the adjacent pie charts (Figure 2C). I would recommend to be consistent between the two representations.

– At line 139, the authors write, concerning R. pumilio SCN activity "hyperpolarized-silent neurons only appearing at night, and the daytime state being characterized by firing and depolarized cells, indicating a time-of-day control on these cellular electrical states". Has this being reported also in the mouse brain? If so, I would state it clearly.

– In Figure 2D and 2F, Mann-Whitney test was used. Why not a t-test given the apparent data distribution?

– In Figure 2, R input (F) does not correlate with RMP and SFR measurements (D and E), a feature considered unusual in the discussion (line 457), as it differentiates R. pumilio electrophysiological properties from the mouse ones. What does this mean biologically? Is that possible that these measures do not correlate always with each other, and if so, in which context can this decoupling occur?

– Line 502: how can the increased excitation of SCN neurons during the day contributes to the health and wellbeing of both diurnal and nocturnal species?

– What are the consequences of the non-correlation between R input and RMP, and the delay-to-fire activity in R. pumilio neurons? As specified in the public comments, I ask the authors to better link the peculiarities identified with the biology leading to a different temporal niche. How can these peculiarities help?

– As KChIP and DPLP have been suggested as candidates to explain the differences based on the electrical properties identified in a mathematical model, it would be interesting to get a biological confirmation at any level (i.e. showing the differences in the expression pattern along the day night cycle in mouse and R. pumilio of these proteins or their transcripts?). The strategy proposed in the brackets represents just a suggestion, and I am open on the use of different strategies: what I would like to see is a biological validation of the model, as currently too much of the conclusions is based on the outcome of purely in silico analysis.

– In general, the comparison before data and model profiles provides several clear differences (Figure 4 (B and C) and Figure 5 A, Figure 6 A´, J, N, Figure S2). I invite the authors to be clearer in the text about the limitations of the modelling approach, and the circumstances where the modelling does not recapitulate well the features observed in the data.

*Reviewer #2 (Recommendations for the authors):*

This study shows circadian behavior in neurons of the SCN in a diurnal animal, complementing the majority of work published on the topic from nocturnal animals. It combines experiments and mathematical modeling, and both aspects are nicely done. A novel finding is the delay-to-fire behavior observed in some neurons that it not seen in studies from nocturnal animals.

Both the experiments and modeling simulations performed in this study are strong, as is the writing (though see below). I do have two major points.

1. A major finding is the delay-to-spike behavior, which distinguishes the electrical behavior of some neurons in this study from those published previously on nocturnal animals. The use of modeling to show that this behavior is likely due to A-current is nicely done, and is a good use of modeling. However, many readers will wonder why the authors did not use the A-type channel blocker 4-AP to demonstrate that when the A-type channels are blocked the delay-to-spike goes away. This should be easily done, and would strengthen one of the main findings of the study.

2. I found most of the paper to be very well written, however, I did not find the same with the Methods section, in particular the section on the data assimilation algorithm. This section is very hard to follow, and is clearly an extraction of text from a dissertation. There are also words and symbols that are undefined. I doubt this long section will be of aid to anyone who does not know the technique, and for those who do it is probably not necessary. I suggest that it be totally rewritten with a focus on clarity, and it would be helpful if all the authors provided feedback. Either that or delete it altogether and simply cite some relevant papers.

*Reviewer #3 (Recommendations for the authors):*

A key question in circadian neurobiology is how the molecular circadian clock regulates the circadian rhythm of suprachiasmatic nucleus neurons. In the current manuscript, Bano-Otalora et al., have combined two novel models to provide insights into the ionic mechanisms that regulate the action potential firing pattern of SCN neurons. The animal model is Rhabdomys pumilio, a rodent species with a pattern of behavioral activity timing different from the nocturnal rodents generally used in circadian research. The second novel component of the manuscript uses a Data Assimilation modeling approach, which allows for ion current parameters to be estimated from the current clamp. The model strategy is potentially an important new computational tool to identify the functional properties underlying neuronal activity. The manuscript will be improved with additional experimental details, improved statistical analyses, and improved integration between the experimental data and the model predictions.

1. Dewey and Dawson (1976) observed and quote Choate (1972) also observed that the Rhabdomys were active at night and their activity should be termed diurnal-crepuscular. They also noted that Rhabdomys were more active at night under laboratory conditions, which should be considered in the discussion of the results.

2. The time frame over which the day and night recordings were made should be added. These data are necessary given the increase in activity observed in Rhabdomys at the light-dark transition and the anticipatory activity observed before lights-on.

3. Line 75 – should clarify that the spontaneous activity referred to is action potential firing.

4. Lines 128-136 – "four spontaneous excitability states" were revealed, but only three are listed – resting at moderate RMPs, severely depolarized or hyperexcitable, hyperpolarized silent.

5. Lines 146-151 – Did the depolarized neurons that were tested with a series of depolarizing and hyperpolarizing steps represent the n = 6 of the group described in the preceding paragraph.

6. Lines 156-157 – Multiple neurons were recorded from a single slice (111 neurons from 8 animals). The data from each neuron is not an independent observation, and a "nested" statistical test should be used.

7. Lines 266-229 – Jackson and Bean used action potential waveforms to study how different currents interact to produce action potentials.

8. Several experimental paradigms were used to test the physiological properties of the SCN neurons. The results of a given protocol were grouped into neurons with specific properties. There is no description of how these properties overlap. For example, there were a group of non-adapting neurons (n = 21) and a group of adapting neurons (n = 81). Following a hyperpolarizing pulse, a group of neurons (n = 69) resumed regular firing while a second group (n = 34) showed a low-threshold spike and delayed action potential firing. How many of the non-adapting neurons were Type A neurons, and how many were Type B neurons? Was the model able to replicate this variability of responses?

9. The H-current is a hyperpolarizing activated current that produces a membrane depolarization. The current clamp traces are shown in Figure 3, the H-current-induced sag plateaus, which suggest that another hyperpolarizing ion current is turning on. In the model currents shown in Figure S3, the calcium current in the Rhabdomys appears to activate faster, be more significant, and last longer than in the mouse SCN model. Could a calcium-activated potassium channel produce the plateau observed in Figure 3?

10. Was a correction made for the liquid junction potential in reporting the membrane potential values?

11. How many traces from how many neurons were used to generate the model parameters? How were these data chosen?

[Editors' note: further revisions were suggested prior to acceptance, as described below.]

Thank you for resubmitting your work entitled "Daily electrical activity in the master circadian clock of a diurnal mammal" for further consideration by *eLife*. Your revised article has been evaluated by the original three reviewers, as well as Catherine Dulac (Senior Editor) and a Reviewing Editor.

All three reviewers very positively noted the significant improvements of the manuscript. However, there are two points that we would nevertheless still ask you to clarify:

1. Lines 546-557 – The authors write that they observed a decoupling between membrane potential and membrane resistance with lower resistance during the day than at night. However, the data shown in Figure 2F shows a nonsignificant difference between the day and night input (see lines 177-183). Furthermore, the membrane potential and input resistance are "decoupled" as the difference in mice and Rhabdomys could reflect the activity and magnitude of different conductances.

2. Based on the description in the methods, it appears that the authors compensated for the electrode offset potential, not the liquid junction potential (see Neher Methods Enzymol. 207:123-131, 1992).

---

## [Author Response]

The reviewers have discussed their reviews with one another, and the Reviewing Editor has drafted this to help you prepare a revised submission.While we think that your study is already quite strong, several important issues have been raised by each of the reviewers. We think that all of them are reasonable and valuable and hence critical to appropriately address. Thus, you don't find a separate summary list here, but please check and respond to each point in the three "Recommendation for the authors" sections.Reviewer #1 (Recommendations for the authors):[…]Overall, I consider this work worthy of publication in eLife, yet I ask the authors to clarify/address the following points:– Why do rodents have diurnal peaks of spontaneous electrical activity (and in general SCN more exited) irrespectively from their diurnal or nocturnal niche? I know this is not the main aim of the work, but I think that the clarity of the entire manuscript will certainly benefit from specifying this. Especially for non experts, this omission generates expectations that are not fulfilled by the findings provided, partially devaluating the work itself.

We thank the reviewer for highlighting this very interesting point, but sadly this is still an outstanding question in the circadian field. That is, how the SCN orchestrates the oppositely phased rhythms of activity in diurnal and nocturnal species: are there any differences in the overall SCN function? Does the SCN have any differential mechanisms to integrate external and internal inputs? Are circadian signals coming from the SCN interpreted in an opposite manner in downstream target areas in diurnal and nocturnal mammals? For example, see (Smale et al., 2003, *J Biol Rhythms*; Smale et al., 2008, *Biological Rhythm Research*).

SCN functions as a clock that tells the rest of the brain and body when it is daytime or nighttime. There are some aspects of SCN-controlled physiology that are invariably locked to day and night e.g. melatonin is high at night in both day- and night-active animals. However, other features need to be more flexible depending on environmental conditions e.g. activity/feeding where rodents may have to balance risk of predation with timing of food availability and/or other metabolic demands. Hence, while in most cases we do not yet know the mechanism that dictate temporal niche in detail, they are thoughts to primarily reside outside of SCN (e.g. involving sign-inverting interneurons in SCN target regions; Kalsbeek et al., 2008, *Eur J Neurosci*).

However, aspects that might be more likely to differ between diurnal and nocturnal rodents relate to responses to environmental inputs, e.g., day-active animals are likely to receive more light (the main excitatory input to the SCN) than nocturnal species (Yan et al.,2020, *Eur J Neurosci*). On the other hand, behavioural feedback to the clock from arousal and wakefulness occurs at different time of day depending on the temporal niche that an animal occupies (Hughes and Piggins, 2012, *Prog Brain Res*). So, such sensory signals might have opposite effects. It is therefore possible that SCN neurophysiology in diurnal species is altered to adapt neuronal function to the specific demands of the animal’s temporal niche preference. Methods used so far to investigate SCN neurophysiology in diurnal species are limited. That is, extracellular recording only reveals the daily variation in SCN neuronal population activity but offers no understanding of the electrophysiological mechanisms involved or the electrical properties of single neurons, particularly on how they respond to inputs.

We agree that specifying this in the introduction would provide clarity for non-experts. As such we have extended the introduction to state our motivation to perform whole-cell recordings to explore the possibility that at least some differences exist at single-cell level which may serve to drive and/or accommodate SCN function to diurnal lifestyles (Please, see lines 72, 80-94).

We hope that by specifying this offers clarity.

– I personally found the description of the aims and the impact too hasty and vague. For example, knowing already that the overall dynamics are preserved between nocturnal and diurnal species (SCN neurons are overall more exited during the day), why did the authors hypothesize a detailed characterization of the whole-cell electrophysiological dynamics could unravel the mysteries of the accommodation of mammalian diurnal niches? I suggest to be more specific since the very beginning on why this approach can help to understand how a diurnal niche could have been accomodated.

We apologize if we have omitted important links for the reader here. As mentioned above, the information provided by extracellular recordings is limited, revealing only the daily variation in SCN neuronal population activity, but offers no understanding of the electrophysiological mechanisms involved or the electrical properties of single neurons (intrinsic mechanisms and neuronal response to inputs), which can be obtained by whole-cell recording.

Therefore, our first aim was to provide a detailed description of the SCN electrical landscape in this diurnal species at single cell level. Second to test SCN neuron responsiveness to inputs, as the nature/timing of environmental and internal inputs to the SCN differs in animals occupying different temporal niches. Therefore, we wanted to investigate whether aspects of the evoked neurophysiological activity in the diurnal SCN are optimised to accommodate SCN function to a diurnal temporal niche. Although the SCN clock adopts the same phase in nocturnal and diurnal species, there are important differences in key inputs responsible for setting the phase and amplitude of this clock according to temporal niche. Thus, diurnal animals are exposed to more bright light during the day, and the reinforcing effects of alertness and activity occur at daytime rather than nighttime.

We hope that changes to the introduction raise these points and provide a clear justification for exploring SCN neurophysiology in diurnal species.

– In line with the last comment, I would like the authors to be more open on the current hypothesis suggested to explain diurnal/nocturnal niche accommodation. Given the literature, probably differences in the electrophysiological dynamics is not the only one. Then they could justify why this is the most likely to them, and thus it deserves to be investigated in depth. To do this, I would start asking: how can species adapt to diurnal or nocturnal niches?

We hope that by addressing the above comments we have also appropriately responded to this one.

As we currently have no understanding of the mechanisms responsible for diurnal/nocturnal niche accommodation, we can only stipulate in line with the literature that the some mechanisms may be in the SCN, downstream from the SCN, or both. We totally agree with the reviewer’ comment in the public review that the first whole-cell recording of SCN neurons from a diurnal species represents a necessary step to better understand the neurophysiological scenario which would potentially accommodate a diurnal niche. We have edited the text in the introduction (lines 80-94) to this effect and provided references.

– In the main text, why VIP, AVP and GRP peptides have been selected is not clear. Only in Figure 1 legend the authors refer to them as "main SCN neuropeptides". I would suggest the authors to better communicate the choice of these neuropeptides in the main text, perhaps providing also one sentence about their function (i.e. circadian regulation, sleep/wake, etc…).

We agree and thank the reviewer for this good suggestion. We have altered the text accordingly and provided references in case the reader would like to follow up on this (Lines 125 to 127).

– Considering Figure 2, it is not clear to me why the authors first described the spontaneous excitability states in Figure 2B (5 states), and then they report only 3 states in the adjacent pie charts (Figure 2C). I would recommend to be consistent between the two representations.

We thank the reviewer for this observation and we apologize for this oversight. We have now clarified this by relabelling Figure 2B and C to indicate that Depolarised-silent and DLAMO cells are grouped together, and same applies for Slow and Fast-firing cells. We have also appropriately edited the text (line 152 and Figure 2 legend) to reflect this.

– At line 139, the authors write, concerning R. pumilio SCN activity "hyperpolarized-silent neurons only appearing at night, and the daytime state being characterized by firing and depolarized cells, indicating a time-of-day control on these cellular electrical states". Has this being reported also in the mouse brain? If so, I would state it clearly.

We thank the reviewer for stating this. Yes, it has, and we have now accordingly stated and referenced this (lines 164-165).

– In Figure 2D and 2F, Mann-Whitney test was used. Why not a t-test given the apparent data distribution?

We appreciate this comment. Normality was initially tested using the Shapiro-Wilk test before performing Mann-Whitney U Test. However, following comment 6 from reviewer 3, since multiple neurons were recorded from a single slice (a total of 111 neurons from 8 animals), we have now analysed electrophysiological data (RMP, SFR and Rinput) using a multilevel mixed-effects linear model that includes the slice that each cell was recorded from as a random effect and the time-of-day as fixed effect. Updated information is now included in the manuscript. We are happy to report that this new analysis has not changed any of our main results and conclusions.

– In Figure 2, R input (F) does not correlate with RMP and SFR measurements (D and E), a feature considered unusual in the discussion (line 457), as it differentiates R. pumilio electrophysiological properties from the mouse ones. What does this mean biologically? Is that possible that these measures do not correlate always with each other, and if so, in which context can this decoupling occur?

An increase in cellular input resistance is mostly associated with reduced potassium channel activity which biologically causes the cell to depolarise. Indeed, in the mouse SCN, increased membrane resistance is associated with depolarised RMP during the day, and at night hyperpolarised cells exhibit reduced membrane resistance (Kuhlman and McMahon, 2004, *Eur J Neurosci*; Belle et al., 2009, *Science*). In *R. pumilio,* this correlation is lost with cells exhibiting low membrane resistance during the day and cells at night having high membrane resistance (Figure 2F). We are not aware of studies reporting this decoupling in SCN neurons.

As for the biological significance for this, in neuronal systems membrane resistance can determine how cells respond to inputs, with high membrane resistance amplifying synaptic signals (e.g. Fernandez et al., 2019, *J Neurosci*; Branco et al., 2016, *Cell*). Therefore, we could speculate that this lack of Rinput-RMP correlation may lead to alteration in the way *R.pumilio* SCN neurons respond to inputs during the day and at night, adapting SCN neurophysiology to the demands of diurnal lifestyles.

We thank the reviewer and we have now added this to our discussion. Please, see lines 507-508 and 546-557.

– Line 502: how can the increased excitation of SCN neurons during the day contributes to the health and wellbeing of both diurnal and nocturnal species?

We thank the reviewer for raising this point. Results from a handful of studies, including our own in the *R. pumilio* (see review Ramkisoensing and Meijer, 2015, *Front Neurol* and Bano-Otalora et al., 2021, *Proc Natl Acad Sci U S A*) show that maintaining a good day/night contrast in SCN electrical activity (high activity during the day and low activity at night), produces high amplitude rhythms in the SCN (indicating a more robust circadian clock). This high amplitude rhythm in the SCN is essential for establishing robust rhythmicity in physiology and behaviour, both in diurnal and nocturnal species; while deterioration of body 24h rhythmicity has detrimental effects on health status (e.g. aging, metabolic and sleep disorders, cancer, mood alterations, etc.).

We have included the above references and altered lines 562-564 to share with the reader that “high day/night contrast” in electrical activity is necessary to sustain the robustness of the clock, a contributing factor to promote health and wellbeing.

– What are the consequences of the non-correlation between R input and RMP, and the delay-to-fire activity in R. pumilio neurons? As specified in the public comments, I ask the authors to better link the peculiarities identified with the biology leading to a different temporal niche. How can these peculiarities help?

We thank the reviewer for this comment. As we instigated in our replies to the above comments, and in the section “Putative functional significance” of the discussion, diurnal lifestyle implies that *animals are exposed to daytime light (the main excitatory input to the SCN) to an extent that nocturnal species are not (for example exposure to high light intensity for longer duration).* As light acts as the main excitatory input to the SCN, it is expected that diurnal species received enhanced excitatory inputs during the day. *The ability of I_A_ to reduce the impact of such excitatory inputs would apply an appropriate ‘brake’ on daytime activity of the SCN.*

By contrast, the intrinsic suppression of SCN electrical activity in nocturnal species at night is augmented/supported by inhibitory inputs associated with activity and arousal at this circadian phase (van Oosterhout et al., 2012). As activity in diurnal animals occurs predominantly during the day, such inhibitory inputs are presumably reduced. The delay to fire and biophysical properties of I_A_ channels could provide an opportunity for the *R. pumilio* SCN to compensate for the reduction in inhibitory inputs at night. Accordingly, our modelling favours the interpretation that I_A_ acts to amplify suppressive signals at night to maintain the low electrical activity in the SCN at this time of day (blockade or deletion of I_A_ currents cause excitation in SCN neurons (Granados-Fuentes et al., 2012, *J Neurosci*; Hermanstyne et al., 2017, *eNeuro*)).

We have added the paragraph below in the discussion, discussing a potential consequence for the decoupling between cellular RMP and membrane input resistance in *R.pumilio* SCN. It reads:

“In the mouse SCN, increased membrane resistance is associated with depolarised RMP during the day, and at night hyperpolarised cells exhibit reduced membrane resistance (Kuhlman and McMahon, 2004; Belle et al., 2009). This decoupling in the R.pumilio resulted from cells exhibiting low membrane resistance during the day and neurons at night having high membrane resistance (Figure 2F). In neuronal systems, membrane resistance can determine how cells respond to inputs, with high membrane resistance amplifying synaptic signals (e.g. (Branco et al., 2016; Fernandez et al., 2019)). The high membrane resistance during the night therefore may provide additional cellular mechanism to amplify inhibitory signals. During the day, depolarised cells with low membrane resistance would be less sensitive to excitatory inputs, thereby supporting the daytime “brake” in extrinsic excitability”.

– As KChIP and DPLP have been suggested as candidates to explain the differences based on the electrical properties identified in a mathematical model, it would be interesting to get a biological confirmation at any level (i.e. showing the differences in the expression pattern along the day night cycle in mouse and R. pumilio of these proteins or their transcripts?). The strategy proposed in the brackets represents just a suggestion, and I am open on the use of different strategies: what I would like to see is a biological validation of the model, as currently too much of the conclusions is based on the outcome of purely in silico analysis.

We thank the reviewer for raising this and we now provide pharmacological confirmation that transient subthreshold A-type potassium channels are the primary determinant of this delay response, as predicted by our modelling. Reviewer 2 also suggested this experiment and we are happy to confirm our model’s prediction.

Showing a differential expression of KChIp and DPLP between the mouse and *R.pumilio* is another very interesting and excellent idea. However, we feel that this is beyond the scope of this manuscript, especially as the lack of a genome sequence for *R. pumilio* makes this a substantial undertaking.

– In general, the comparison before data and model profiles provides several clear differences (Figure 4 (B and C) and Figure 5 A, Figure 6 A´, J, N, Figure S2). I invite the authors to be clearer in the text about the limitations of the modelling approach, and the circumstances where the modelling does not recapitulate well the features observed in the data.

We thank the reviewer for raising this valuable suggestion. As requested, we have now included the following paragraph detailing some of the limitations of the model in the Discussion section (Lines 566 to 587).

“Computational modelling approach and considerations

Mammalian neurons, including SCN neurons, possess a wide array of ion channels that contribute to membrane excitability and action potential generation. Our computational model of R. pumilio SCN neurons incorporates several of the ionic currents that have been observed in mouse SCN, but some currents are not represented in the model. For example, the large-conductance calcium-activated potassium (BK) channel is known to play a role in circadian variation of SCN excitability (Whitt et al., 2016), but we have not included it in our model due to the added complexity involved in modelling intracellular calcium dynamics. Furthermore, the currents that are in the model do not distinguish among the different subtypes of current that exist for each ion. For example, the model contains a single inward calcium current, rather than separate L-, N-, P/Q-, and R-type calcium currents that have distinct activation/inactivation kinetics and are known to be present in SCN neurons (McNally et al., 2020). For some currents that are in the model we also make simplifying assumptions to reduce the number of parameters that need to be estimated. For example, since it is known that the inward sodium current in SCN neurons activates very rapidly, in our model we assume that it activates instantaneously so that we do not have to include parameters associated with the time constant for activation. The lack of a voltage-dependent time constant for sodium activation in our model may explain the subtle difference in the shape of the upstroke of the action potentials in the model compared to the data (Figure 4C). In addition, since our model is deterministic, it does not capture the irregularity in spike timing or the small voltage fluctuations that are present in the recordings due to ion channel noise or synaptic input (e.g. see Figure 4—figure supplement 2).”

Reviewer #2 (Recommendations for the authors):Both the experiments and modeling simulations performed in this study are strong, as is the writing (though see below). I do have two major points.1. A major finding is the delay-to-spike behavior, which distinguishes the electrical behavior of some neurons in this study from those published previously on nocturnal animals. The use of modeling to show that this behavior is likely due to A-current is nicely done, and is a good use of modeling. However, many readers will wonder why the authors did not use the A-type channel blocker 4-AP to demonstrate that when the A-type channels are blocked the delay-to-spike goes away. This should be easily done, and would strengthen one of the main findings of the study.

We thank the reviewer for these very positive comments, and for raising this important point. We have performed the necessary pharmacology experiments with 4-AP to confirm our model’s prediction that transient subthreshold A-type potassium channels are indeed responsible for the delay-to-spike behaviour (Figure 6, new panels H and I). We have altered the result section (lines 353-359, 367-368, 375,) as well as the abstract (line 55), discussion (Line 449) and methods (lines 637-639, 702-704) to reflect this new addition. We agree that this strengthens one of the main findings of the study.

2. I found most of the paper to be very well written, however, I did not find the same with the Methods section, in particular the section on the data assimilation algorithm. This section is very hard to follow, and is clearly an extraction of text from a dissertation. There are also words and symbols that are undefined. I doubt this long section will be of aid to anyone who does not know the technique, and for those who do it is probably not necessary. I suggest that it be totally rewritten with a focus on clarity, and it would be helpful if all the authors provided feedback. Either that or delete it altogether and simply cite some relevant papers.

We agree with the reviewer and have made the following changes to address this comment. First, we have moved the details of the data assimilation algorithm to an appendix (Appendix 1), retaining only the two sections most relevant to the majority of readers (Model Estimation Strategy and Conductance-Based Model) in the main manuscript. Second, we edited all sections of the Methods (both those in main manuscript and those in the appendix 1) for clarity.

Reviewer #3 (Recommendations for the authors):A key question in circadian neurobiology is how the molecular circadian clock regulates the circadian rhythm of suprachiasmatic nucleus neurons. In the current manuscript, Bano-Otalora et al., have combined two novel models to provide insights into the ionic mechanisms that regulate the action potential firing pattern of SCN neurons. The animal model is Rhabdomys pumilio, a rodent species with a pattern of behavioral activity timing different from the nocturnal rodents generally used in circadian research. The second novel component of the manuscript uses a Data Assimilation modeling approach, which allows for ion current parameters to be estimated from the current clamp. The model strategy is potentially an important new computational tool to identify the functional properties underlying neuronal activity. The manuscript will be improved with additional experimental details, improved statistical analyses, and improved integration between the experimental data and the model predictions.1. Dewey and Dawson (1976) observed and quote Choate (1972) also observed that the Rhabdomys were active at night and their activity should be termed diurnal-crepuscular. They also noted that Rhabdomys were more active at night under laboratory conditions, which should be considered in the discussion of the results.

We thank the reviewer for raising this point. In our hands, we have found *R.pumilio* to be strongly diurnal under laboratory conditions. Please see figure 1 in our recently published work (Bano-Otalora et al., 2021, *Proc Natl Acad Sci U S A*). This also reflects more recent descriptions of *R.pumilio* in the field and the experience of other laboratory colonies.

In line with Dewey and Dawson observations, we also see an increase in activity in anticipation to lights on. However, in our case animals sustained high activity levels across the light period, except during a “siesta” period in the middle of their light phase, and very low activity during the night (usually coinciding with the animal drinking water or changing position). Ecological and genetic differences (although striped mice have been classified as a single species, karyotypic and genetic evidence suggests that there are at least two species, *Rhabdomys pumilio* and *R. dilectus*, Mallarino et al., 2018, *Current Biol*) as well as environmental housing conditions may account for the behavioural differences observed across the different studies. However, the designation of *R. pumilio* as a reliably day-active species is supported by other aspects of its biology, particularly its visual system which has several adaptations (low rod content, UV blocking lens) that are typical of animals relying on daytime vision (Allen et al., 2020, *J Exp Biol*).

We agree that this more complete description of *R.pumilio* phenotype would be helpful for the reader and have adjusted the introduction accordingly (line 98-99, 101-103). We have also extended the details of the lighting conditions we kept our animals under (lines 608-611), and included details for the origins of our animal colony in Manchester (lines 613-614).

2. The time frame over which the day and night recordings were made should be added. These data are necessary given the increase in activity observed in Rhabdomys at the light-dark transition and the anticipatory activity observed before lights-on.

We thank the reviewer for this comment and we apologize for not including this information before. Time frame for the recordings (day (ZT4 to ZT12) and night (ZT13 to ZT22)) is now included in the method section, lines 641.

3. Line 75 – should clarify that the spontaneous activity referred to is action potential firing.

We thank the reviewer for identifying this and we have now clarified this (Line 77).

4. Lines 128-136 – "four spontaneous excitability states" were revealed, but only three are listed – resting at moderate RMPs, severely depolarized or hyperexcitable, hyperpolarized silent.

We thank the reviewer for this observation, and a similar point is also raised by reviewer 1. We apologize for this oversight. We have now clarified this by relabelling Figure 2B and C to indicate that Depolarised-silent and DLAMO cells are grouped together, and same applies for Slow and Fast-firing cells. We have also appropriately edited the text (line 152 and Figure 2 legend) to reflect this.

5. Lines 146-151 – Did the depolarized neurons that were tested with a series of depolarizing and hyperpolarizing steps represent the n = 6 of the group described in the preceding paragraph.

We apologize for not having specified this information in the text. We tested the progressive hyperpolarizing steps in two depolarized cells. This is now indicated in the text (line 170).

6. Lines 156-157 – Multiple neurons were recorded from a single slice (111 neurons from 8 animals). The data from each neuron is not an independent observation, and a "nested" statistical test should be used.

We agree with the reviewer. We have now rerun our analysis using this approach. We are happy to report that this has not changed any of our results. We have altered the result section and figure 2 legend appropriately and reported the statistical test performed in the methods (Lines 699-702), to read:

“Since multiple neurons were recorded from a single slice (a total of 111 neurons from 8 animals), electrophysiological data (RMP, SFR and R_input_) were compared using a multilevel mixed-effects linear model that included the slice that each cell was recorded from as a random effect and the time-of-day as fixed effect.”

7. Lines 266-229 – Jackson and Bean used action potential waveforms to study how different currents interact to produce action potentials.

We thank the reviewer and based on his comment, we have removed the phrase “information that could never be obtained experimentally since current-clamp and voltage-clamp cannot be simultaneously performed”.

8. Several experimental paradigms were used to test the physiological properties of the SCN neurons. The results of a given protocol were grouped into neurons with specific properties. There is no description of how these properties overlap. For example, there were a group of non-adapting neurons (n = 21) and a group of adapting neurons (n = 81). Following a hyperpolarizing pulse, a group of neurons (n = 69) resumed regular firing while a second group (n = 34) showed a low-threshold spike and delayed action potential firing. How many of the non-adapting neurons were Type A neurons, and how many were Type B neurons? Was the model able to replicate this variability of responses?

We thank the reviewer for raising this important point which we agree will provide useful information for our readers. We have now included the following paragraph in the result section addressing how these neuronal properties overlap:

“analysis of the relationship between cellular responses to depolarizing and hyperpolarizing pulses revealed that non-adapting and adapting neurons exhibited similar proportions of rebound or delay-to-fire behaviours (~34-38% delay and ~62-66% rebound), suggesting that non-adapting or adapting responses in these neurons do not determine firing characteristics to hyperpolarizing pulses.”

This conclusion is supported by our simulations, since models that exhibit delay-to-fire behaviours in response to hyperpolarizing pulses can exhibit either adapting or non-adapting responses to depolarizing pulses (e.g. models shown in Figure 6B and 6O, respectively).

9. The H-current is a hyperpolarizing activated current that produces a membrane depolarization. The current clamp traces are shown in Figure 3, the H-current-induced sag plateaus, which suggest that another hyperpolarizing ion current is turning on. In the model currents shown in Figure S3, the calcium current in the Rhabdomys appears to activate faster, be more significant, and last longer than in the mouse SCN model. Could a calcium-activated potassium channel produce the plateau observed in Figure 3?

We thank the reviewer for such an astute observation. We simulated the model shown in Figure 6B, which has a prominent H-current-induced sag plateau, and plotted the calcium current (I_Ca_). The model predicts that there is indeed calcium current present throughout the hyperpolarizing pulse, with an amplitude of around -40 pA during the plateau. At first this surprised us, since we anticipated that the calcium current would not be highly activated at the hyperpolarized membrane potential of the plateau (around -70 mV). Plotting the gating variable for the calcium channels shows that this is the case, as the activation gating variable mCa is quite small during the pulse. On the other hand, the hyperpolarizing pulse also de-inactivates the calcium channels, as hCa approaches 1 shortly after the pulse begins. Thus, the model does suggest that there could be enough I_Ca_ to potentially turn on a calcium-activated potassium channel. However, we have not included a calcium-activated potassium channel in our model (due to the complexities of modelling intracellular calcium dynamics), so we cannot examine this hypothesis further with the current version of the model. Author response image 1 showing the voltage, I_Ca_, and the gating variables during the hyperpolarizing pulse is included.

10. Was a correction made for the liquid junction potential in reporting the membrane potential values?

*The junction potential between the pipette and extracellular solution was zeroed before establishing a seal and was not further corrected*. This information is now included in the method section (Lines 655-656).

11. How many traces from how many neurons were used to generate the model parameters? How were these data chosen?

We thank the reviewer for raising this, and we apologize for not included this information in the original submission. We have added the following to the caption the Table in Supplementary file 1:

Models were fit to data from 7 different neurons, including a non-adapting cell (Figure 5Ai), an adapting-firing cell (Figure 5Aii), an adapting-silent cell (Figure 5Aiii), two Type-A rebound spiking cells (Figures 4 and 6A), and two Type-B delay cells (Figure 6B and 6O). These cells were chosen for modelling because they are representative of the various responses observed across all the recordings. Each model was fit to data from 6 voltages traces as illustrated in Figure 4—figure supplement 1.

[Editors' note: further revisions were suggested prior to acceptance, as described below.]

All three reviewers very positively noted the significant improvements of the manuscript. However, there are two points that we would nevertheless still ask you to clarify:1. Lines 546-557 – The authors write that they observed a decoupling between membrane potential and membrane resistance with lower resistance during the day than at night. However, the data shown in Figure 2F shows a nonsignificant difference between the day and night input (see lines 177-183). Furthermore, the membrane potential and input resistance are "decoupled" as the difference in mice and Rhabdomys could reflect the activity and magnitude of different conductances.

We thank the editors/reviewers for pointing out the lack of clarity in this paragraph. We have revised these lines in the manuscript. It now reads:

“Our results suggest a potential decoupling between cellular RMP and membrane/input resistance (R_input_) in R. pumilio SCN neurons. In the mouse SCN, increased membrane resistance is associated with depolarised RMP during the day, and at night hyperpolarised cells exhibit reduced membrane resistance (Kuhlman and McMahon, 2004; Belle et al., 2009). In the R. pumilio SCN, we found that RMP is more depolarised during the day than at night as in the mouse SCN, but that there is not a significant day-night difference in membrane resistance (Figure 2F) due to a subset of day cells with relatively low R_input_ (despite a depolarized RMP) and a subset of night cells with relatively high R_input_ (despite a hyperpolarized RMP). The decoupling of membrane potential and input resistance in the R. pumilio SCN could reflect the activity and magnitude of different conductances in this species compared to mouse SCN. In neuronal systems, membrane resistance can determine how cells respond to inputs, with high membrane resistance amplifying synaptic signals (e.g. (Branco et al., 2016; Fernandez et al., 2019)). In R. pumilio, cells with high membrane resistance during the night therefore may provide an additional cellular mechanism to amplify inhibitory signals. During the day, depolarised cells with low membrane resistance would be less sensitive to excitatory inputs, thereby supporting the daytime “brake” in extrinsic excitability”.

2. Based on the description in the methods, it appears that the authors compensated for the electrode offset potential, not the liquid junction potential (see Neher Methods Enzymol. 207:123-131, 1992).

We thank the editors/reviewers, and we have revised the sentence to now read:

“The electrode offset potential was compensated before establishing a seal and the liquid junction potential was not corrected”.